# Convergent mapping of a tremor treatment network

Lukas L. Goede [1,2] ✉, Bassam Al-Fatly [2], Ningfei Li [2], Leon K. Sobesky[3], Bahne H. Bahners [1,4,5], Patricia Zvarova [1,2,6], Clemens Neudorfer [1], Martin Reich [7], Jens Volkmann [7], Chencheng Zhang [8], Vincent J. J. Odekerken[9], Rob M. A. de Bie [9], Ellen F. P. Younger[10], Daniel T. Corp[10,11], Erik H. Middlebrooks[12], Juho Joutsa [11,13], Michiel Dirkx [14], Günther Deuschl[15], Rick C. Helmich[14], Andrea A. Kühn [2], Michael D. Fox [1] & Andreas Horn [1,16,17]

Tremor occurs in various forms across diverse neurological disorders, including Parkinson's disease and essential tremor. While clinically heterogeneous, converging evidence suggests a shared brain network may underlie tremor across conditions. Here, we empirically define such a network using four modalities: lesion locations, atrophy patterns, EMG-fMRI, and deep brain stimulation outcomes. We show that network connectivity robustly explains clinical outcomes in independent cohorts undergoing deep brain stimulation of the subthalamic nucleus for Parkinson's disease and the ventral intermediate nucleus for essential tremor. Maps from each cohort accounted for outcomes in the respective other, supporting a disorder-independent tremor network. A multimodal agreement map revealed consistent substrates in the primary motor cortex and motor cerebellum. To validate the network, we test its predictive power in a third, independent cohort treated with pallidal stimulation for Parkinson's disease. Our findings define a robust, cross-condition tremor network that may guide both invasive and noninvasive neuromodulation strategies.

Tremor is defined as an involuntary, rhythmic oscillatory movement of a body part and represents one of the most common symptoms in movement disorders[1]. It may occur in various forms and as a symptom of different disorders, including physiological tremor, essential tremor (ET), Parkinsonian tremor, dystonic tremor, cerebellar tremor, multiple sclerosis tremor, and others. Symptomatology can be broadly classified into rest, postural, and action tremor, while other and overlapping forms exist. ET, the most prevalent movement disorder, affects over 3% of individuals aged 50 and older, while Parkinson's disease (PD) impacts approximately 0.6% of people older than 45 years[2,3]. Pharmacotherapy can be insufficient, and deep brain stimulation (DBS) has become a well-established treatment option, significantly alleviating tremor and improving clinical scores for both PD

and ET[4,5]. However, DBS is not maximally effective in all patients, tremor can re-occur (habituation), and DBS can induce side effects such as ataxia and dysarthria. There is also a lack of consensus on the exact therapeutic target for tremor and whether different types of tremors require different targets. For example, it has not yet been resolved whether ventral intermediate nucleus of the thalamus (VIM), posterior subthalamic area (PSA), or white matter tracts connecting the thalamus to the cerebellum are the ideal DBS targets for tremor (for recent reviews see refs. 6,7).

Evidence accumulates that the ideal target for neurological symptoms may be a connected brain circuit, not one specific brain region. At first approximation, high-frequency DBS acts as an 'informational lesion' and disrupts pathological activity in specific brain

circuits[8,9]. Indeed, similar if not the same targets can successfully reduce tremor with either ablative lesioning or DBS[10-13]. In rare cases, naturally occurring brain lesions as in form of stroke can also lead to reduced tremor[14-17]. Using a method termed 'lesion network mapping'[18], such lesions have recently been mapped onto a common brain network. This connected network had a peak hub in the VIM, which precisely corresponds to the key DBS target mentioned above[19] and involved primary motor cortex and motor cerebellum. In other words, naturally occurring lesions that led to the cessation of tremor were almost always strongly connected to these specific brain regions. Alongside numerous historical and modern evidence[10,20], this work further supported the hypothesis that the therapeutic target for tremor reduction is a connected brain circuit, not one specific brain region, and that DBS targeting this circuit may optimize tremor improvement[21].

Studies using a related approach, known as DBS network mapping, have identified relevant networks associated with overall clinical improvement in both PD and ET patients[22,23]. While the VIM is the most commonly targeted region for ET patients, GPi and STN are the preferred targets for PD patients—both of which can also alleviate PD-related tremor[4]. Within PD, DBS network mapping has already been employed to compare different DBS targets, which identified a common PD response network[24]. This finding helps to explain why DBS targeting different regions within PD, specifically the STN and the GPi, may yield similar therapeutic effects[24]. In parallel, this and similar research has shifted focus from disease-specific to symptom-specific networks, as demonstrated exemplarily in studies on depression or PD[25,26].

For tremor specifically, DBS network mapping identified a set of brain regions[23,24] very similar to those found in lesion network mapping studies reported earlier[19], primarily involving the primary motor cortex and motor parts of the cerebellum. Moreover, a recent study that subjected atrophy patterns to network mapping analyses yet again identified a key hub in matching (motor) regions of the cerebellum[27]. Finally, studies using concurrent electromyography (EMG) and functional MRI (fMRI) have also explored tremor-related activity in the brain and once more identified a shared set of connections between thalamus, cerebellum, and primary motor cortex with tremor[28,29].

These findings converge on the notion that the optimal target for tremor relief is not a single brain region but rather a network that includes shared hubs while allowing for disorder-specific variations. Therefore, comparing and integrating results from lesion network mapping, fMRI, atrophy, and DBS network mapping could yield a unified definition of a tremor treatment network. However, no study has yet analyzed converging evidence from these maps and set the resulting network pattern into relationship with tremor outcomes across disorders (ET and PD) or across different DBS targets (VIM, STN, and GPi). Here, we aim at addressing this gap by first investigating whether networks identified in lesion network mapping, EMG-fMRI, and atrophy network mapping studies could explain variance in clinical improvements across large cohorts of patients who underwent DBS surgery for PD and ET targeting the STN and VIM. Second, we develop a data-driven tremor network using these DBS cohorts to determine whether they independently replicate the same tremor treatment network across disorders and DBS targets. Third, we integrate these findings with a priori results from lesion network mapping, fMRI, and atrophy network mapping to generate a multimodally informed tremor network. Finally, we test whether this network may account for improvements in tremor in a further large cohort of PD patients that underwent DBS to the GPi, as a third DBS target.

## Results
### Clinical cohorts and tremor improvement
The study sample first included 65 hemispheres from $N = 47$ PD patients who had undergone STN-DBS, with corresponding contralateral tremor improvements. The mean ($\pm$ SD) baseline MDS-UPDRS-III tremor score was 4.69 ($\pm$ 1.97) points, and the mean ($\pm$ SD) improvement post-DBS was by 4.10 ($\pm$ 1.75) points (87.4% reduction). In addition, second, a VIM-DBS cohort operated in Berlin was analyzed, comprising 72 hemispheres from $N = 36$ ET patients with corresponding hemi scores. The mean baseline Fahn-Tolosa-Marin Clinical Rating Scale for Tremor (FTM)-Tremor-Score was 13.85 ($\pm$ 4.47) points, and the mean improvement was 9.28 ($\pm$ 4.78) points (67.0%). Third, the ET comparison cohort from Jacksonville, Florida, for the VIM group, had a mean baseline score in the total FTM-Tremor-Score of 41.47 ($\pm$ 10.86) points and a mean improvement of 20.52 ($\pm$ 8.16) points (50%). Finally, fourth, the out-of-sample test dataset included 31 hemispheres from $N = 22$ PD patients who received GPi-DBS. The mean baseline MDS-UPDRS-III tremor score for this group was 5.19 ($\pm$ 2.20) points, with a mean improvement of 4.68 ($\pm$ 2.61) points (90.1%).

### Comparison of lesion-, atrophy-, and EMG-fMRI- maps
Similarity coefficients between the published a-priori tremor-relief map that was based on lesion network mapping of eleven cases that had tremor alleviation following a stroke[19] and the connectivity maps derived by volumes of activated tissue (VTA) around the DBS electrodes significantly correlated with empirical clinical improvements of DBS patients (rho(135) = 0.25, $p = 0.003$, 95% CI [0.09, 0.40]; Fig. 1). This was done after regressing out cohort grouping (STN-DBS vs. VIM-DBS) from connectivity estimates. When instead accounting for cohort grouping in a joint linear model, results remained significant: $R^2 = 0.13$, $F = 10.27$, $p = 10^{-4}$. This finding validated the a priori lesion map using DBS and suggests that a therapeutic target from tremor is a brain circuit that shares topography across multiple lesion locations, multiple DBS sites, and multiple underlying diagnoses to alleviate tremor. When repeating the analysis for each DBS cohort alone, clinical improvements correlated positively with connectivity to the lesion network map in both cohorts, but this association was only significant for VIM-DBS for ET (rho(70) = 0.29, $p = 0.015$, 95% CI [0.07, 0.49]) and not for STN-DBS for PD (rho(63) = 0.15, $p = 0.211$, 95% CI [−0.09, 0.38]).

A similar analysis was conducted with a published network map derived from atrophy network mapping in ET patients[27]. The map included a network hub that was both sensitive and specific to ET. While connectivity between our DBS sites to this published map did not significantly correlate with clinical improvements, a positive association was found (rho(135) = 0.11, $p = 0.211$, 95% CI [−0.07, 0.26]; not shown).

A third analysis was conducted with a published network map that had been established using a concurrent EMG-fMRI experiment on 22 patients with PD, which led to the influential 'dimmer-switch' model of tremor[28]. This analysis also revealed a significant association between connectivity to the fMRI tremor activity map and clinical improvements (rho(135) = 0.22, $p = 0.013$, 95% CI [0.06, 0.38]; Fig. 2). As above, this analysis was carried out after regressing out cohort grouping (STN-DBS vs. VIM-DBS) from connectivity estimates. When instead accounting for cohort grouping in a joint linear model, results remained significant: $R^2 = 0.13$, $F = 9.69$, $p = 10^{-3}$. When repeating the analysis for each DBS cohort alone, clinical improvements correlated positively with connectivity to the EMG-fMRI map in both cohorts, but this association was only significant for STN-DBS for PD (rho(63) = 0.24, $p = 0.046$, 95% CI [0.00, 0.46]) and not for VIM-DBS for ET (rho(70) = 0.21, $p = 0.074$, 95% CI [−0.03, 0.42]).

These three sources of information (lesions, atrophy patterns, and fMRI results) commonly pointed to a network that involved primary motor cortex and specific parts of the motor cerebellum. Connectivity from each DBS site to the respective three maps was positively associated with clinical improvements following DBS in two targets and disorders (relationship n.s. for the atrophy map). Next, we wanted to investigate the DBS data in a data-driven approach, to test whether these data would again point to a similar network when analyzed

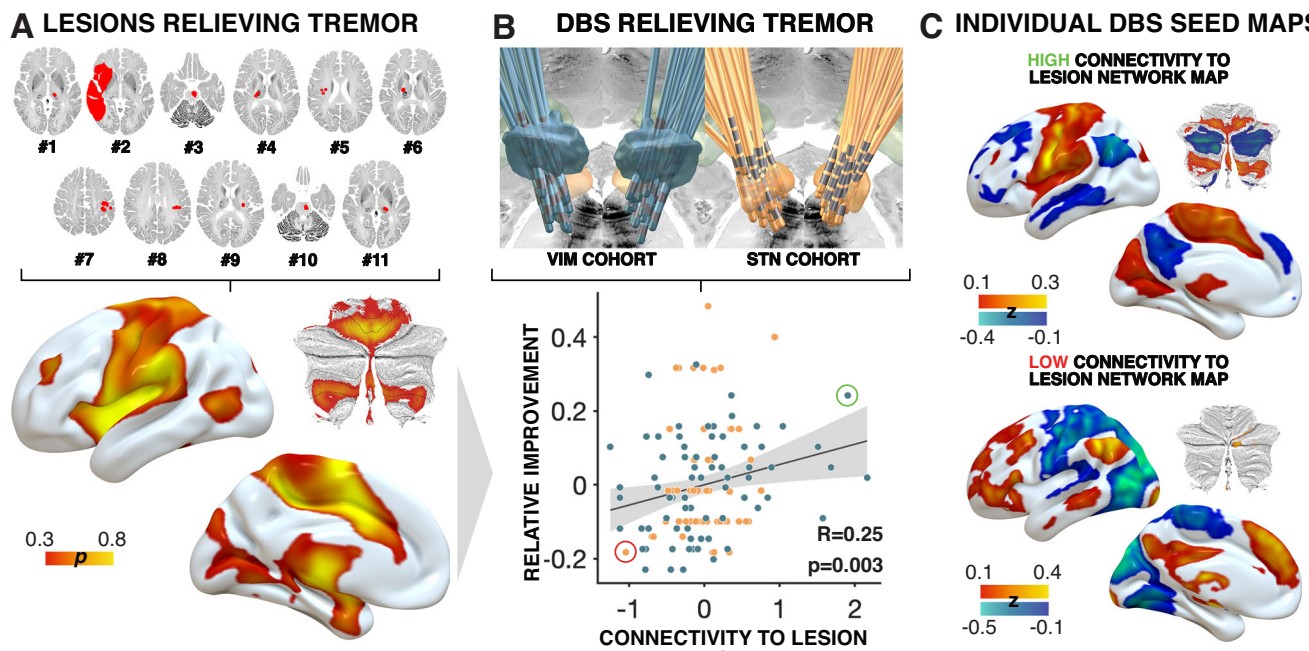

**Fig. 1 | Lesion-derived *tremor-relief map* connectivity correlates with clinical improvement in deep brain stimulation (DBS) cohorts. A** The panel illustrates 11 published cases of patients who experienced tremor alleviation following beneficial lesions due to stroke[19]. Using lesion network mapping, a tremor-relief network was identified (bottom). Colors represent the probability (p) of voxels associated with tremor relief. **B** Electrode localizations from subjects in the current study, with blue representing VIM patients and orange representing STN patients. The bottom plot shows a positive Spearman correlation (two-sided) between relative clinical improvement (measured as ratios of the maximum achievable score for UPDRS-III tremor scores or FTM) following DBS and each DBS electrode's connectivity to the lesion network map. A cohort regressor was applied to control for potential confounding effects related to differences between patient groups. No adjustment for multiple comparisons was applied. The shaded area indicates the 95% confidence interval. The red circle highlights an example of a volume of activated tissue (VTA) with low connectivity, shown alongside an example with high connectivity (green circle) to the lesion network map in (**C**). Transversal and coronal slices from the BigBrain atlas were used[61]. Cerebellar flatmaps were created using the SUIT toolbox[62]. DBS Deep Brain Stimulation, FTM Fahn-Tolosa-Marin Tremor Rating Scale, GPi Globus pallidus internus, STN Subthalamic nucleus, UPDRS-III Unified Parkinson's Disease Rating Scale, part III, VIM Ventral intermediate nucleus of the thalamus.

independently. Indeed, both the STN-DBS (PD) and VIM-DBS (ET) cohorts independently associated functional connectivity to similar regions with optimal tremor outcomes. Namely, the primary motor cortex (M1), motor cerebellum, and supplementary motor area alongside visual regions positively correlated with tremor improvements (Fig. 3). These models accounted for variance in clinical outcomes within the same sample (STN cohort: $R = 0.46$; VIM cohort: $R = 0.43$). These analyses were circular and meant to demonstrate the degree of fit between model and data, i.e., the maximal possible amount of variance that can be explained by these maps. Due to circularity, we avoid reporting *p*-values of these correlations. However, models remained significant when subjected to fivefold cross-validation, which circumvents circularity (STN cohort: rho(63) = 0.33; $p = 0.004$, [0.00, 0.42]; VIM cohort: rho(70) = 0.35, $p = 0.001$, 95% CI [0.16, 0.56]). In these analyses, for each target, the optimal maps were calculated five times, each time leaving out one fold of patients, and calculated maps were used to estimate outcomes in the respective left-out fold. For results when repeating analyses using additional phenotypical data and replication of VIM-DBS results on an independent cohort from a second center, see supplement.

**DBS cohort comparison**
While data-driven maps from STN- and VIM-DBS cohorts looked similar by visual inspection (Fig. 3), we empirically tested their similarity in two ways. First, we used one of the maps to account for variance in clinical outcomes in the respective other cohort (VIM-DBS map predicts STN-DBS patients: rho(63) = 0.24, $p = 0.049$, 95% CI [0.00, 0.46]; STN-DBS map predicts VIM-DBS patients: rho(70) = 0.31, $p = 0.009$, 95% CI [0.08, 0.50]; Fig. 3A). Second, we tested whether similarities

across the two maps, as expressed by voxel-wise spatial correlations, would be higher than what could be expected by chance. To do so, we permuted outcomes in one cohort and each time compared the similarity of the resulting *R-map* with the respective other (unpermuted) *R-map* to generate null-models of similarities (10,000 iterations). In both directions, similarities of unpermuted *R-maps* were significantly larger than expected by chance (permuting STN-DBS: $p = 0.02$; permuting VIM-DBS: $p = 0.03$; permuting both cohorts at the same time: $p = 0.02$; Fig. 3B).

Despite the two maps being similar, they were not identical. We created an *'agreement map'* that only retained voxels that had the same sign in both maps[24] and tested whether this map would be able to estimate tremor outcomes across both groups, which was the case for circular analysis ($R = 0.41$) and fivefold cross-validation (rho(135) = 0.20, $p = 0.018$, 95% CI [0.04, 0.36]; Fig. 4).

**Convergent tremor map and out-of-sample analysis**
The five maps (two DBS maps and three a priori maps) featured similar profiles across the brain but were not identical. This motivated us to create an even more selective *agreement map* across all five maps using the same concept, i.e., by only retaining voxels that had the same sign across all five maps[24]. Of note, since the atrophy map only defined voxels in the cerebellum, it did not have an influence on the cerebral parts of the map. The resulting multimodally-informed convergent tremor map, illustrated in Fig. 5, may include the most condensed and relevant brain regions associated with tremor alleviation. Specifically, it revealed a strong positive correlation with the primary motor cortex (M1) and motor parts of the cerebellum, as well as visual cortex, and a negative correlation with the dorsolateral prefrontal cortex. To test

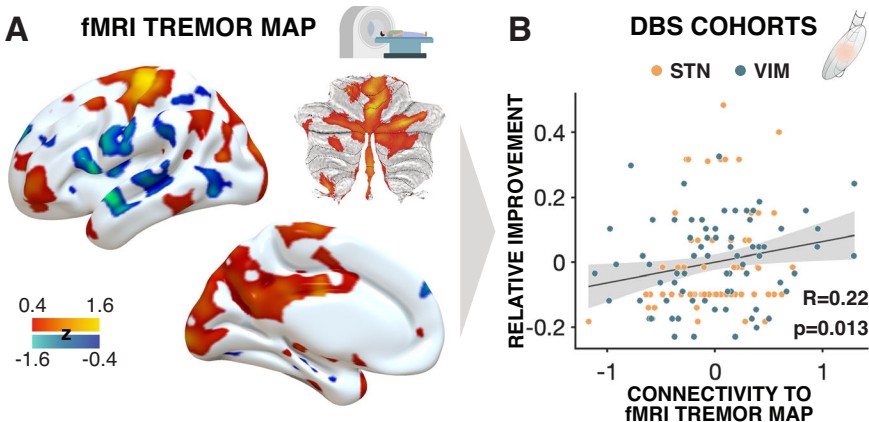

**Fig. 2 | fMRI derived *tremor activity map* connectivity correlates with clinical improvement in deep brain stimulation (DBS) cohorts. A** This panel illustrates tremor-amplitude-related activity within the cerebello-thalamo-cortical circuit, derived from individual concurrent EMG-fMRI data in 22 cases[28]. Colors represent z-scores (z). Cerebellar flatmaps were created using the SUIT toolbox[62]. **B** The plot shows a positive Spearman correlation (two-sided) between relative clinical improvement (measured as ratios of the maximum achievable score for UPDRS-III tremor scores or FTM) following DBS and each DBS electrode's connectivity to the fMRI-derived tremor map. Orange and blue dots represent electrodes from the STN and VIM cohorts, respectively. A cohort regressor was applied to control for potential confounding effects related to differences between patient groups. No adjustment for multiple comparisons was applied. The shaded area indicates the 95% confidence interval. Illustrations were created using Affinity Designer 2 (Serif Ltd., Nottingham, UK). DBS Deep Brain Stimulation, FTM Fahn-Tolosa-Marin Tremor Rating Scale, STN Subthalamic nucleus, UPDRS-III Unified Parkinson's Disease Rating Scale, part III, VIM Ventral intermediate nucleus of the thalamus.

utility of this map, we used it to predict ranks in an independent out-of-sample cohort of PD patients who were operated on at a third DBS target, the GPi. This dataset was deliberately held back from prior analyses to represent a truly independent cohort of patients, that was also yet again different in nature (different DBS target). We deliberately chose to hold out this dataset to test whether the convergent tremor map would generalize to datasets that would be *different in nature*. Connectivity from DBS sites in the GPi cohort significantly correlated with connectivity to the convergent tremor map (rho(29) = 0.45, $p = 0.009$, 95% CI [0.12, 0.70]; Fig. 5) and remained unchanged and significant when repeating the analysis without including the atrophy map (Fig. S7).

To test whether different forms of tremor would map to the same network, we repeated this analysis only using rest vs. action/postural tremor items (Figs. S8 and S9).

## Replication with a disease-matched connectome

Our analyses applied normative connectome data acquired in healthy subjects. While a disease-matched connectome acquired in ET patients was not available, we replicated main results using a disease-matched connectome acquired in PD patients. This confirmed that optimal response networks remained consistent when using a PD connectome, suggesting that results were not substantially biased by the connectome itself. Furthermore, predictive performance improved for the out-of-sample cohort test, yielding an enhanced outcome estimation in the GPi-DBS PD cohort (rho(29) = 0.63, $p = 0.0004$, 95% CI [0.36, 0.81]; Fig. S10).

## Discussion

This study aimed at delineating the neural network underlying tremor suppression through a comprehensive analysis of datasets across various modalities, including DBS network mapping, lesion network mapping, atrophy network mapping, and EMG-fMRI. Our multimodal approach integrated data from different disorders (PD and ET), patient cohorts (DBS, lesions, atrophy patterns, and EMG-fMRI), and different DBS targets (STN, VIM, and GPi). Four main conclusions may be drawn: First, a newly introduced high-resolution connectome could effectively disentangle brain networks associated with clinical improvements, revealing patterns that have not previously been demonstrated in the context of tremor across diseases using fMRI. Second, tremor

appeared to respond to a convergent tremor-suppressing network, independent of the modulating intervention, whether it was a naturally occurring lesion or a 'virtual lesion' induced by DBS. Third, there is potential for symptom-specific neuromodulation strategies which would target the resulting tremor network, irrespective of the underlying disease. Finally, fourth, the identified network was able to explain variance in an independent GPi-DBS cohort, which further indicates generalizability of findings. This dataset was deliberately held out of the primary analyses to constitute an unbiased test sample that was different in nature from the datasets used to characterize the convergent tremor network in the first place. Connectivity between GPi-DBS electrodes with the convergent tremor map accounted for significant amounts of variance, despite no dataset used to create the map involved the GPi.

The idea of cortical and subcortical *network hubs* within a symptom-specific tremor brain network was early described by multiple groups[6], such as the Freiburg school of stereotaxy in the 1950s and 60s[10]. In an analysis of 560 ablation cases between 1950 and 1958, they identified that effective tremor control involved targeting a circuit comprising the cerebellum (specifically the emboliform nucleus and dentate), ventral intermediate nucleus, and primary motor cortex, findings that now have largely been replicated by numerous modern DBS imaging studies[6,7,12,20,30–33]. The lesion-derived tremor relief map published by Joutsa et al.[19], which shows positive connectivity to the VIM and motor cortex (M1), aligns with these findings, highlighting the VIM's crucial role in tremor modulation and underscoring the enduring relevance of these neuroanatomical relationships. However, the improvement of tremor with STN- or GPi-DBS, despite their lack of direct cerebellar input, may imply the involvement of additional mechanisms. Local field potential recordings from STN-DBS have detected high-frequency patterns in the STN during tremor, and increased subthalamic nucleus-cortex, cortico-muscular, and sub-thalamic nucleus-muscle coherence during tremor episodes[34,35]. Potentially, the functional polysynaptic connections identified in the present study may provide the missing link in understanding tremor improvement across different targets and modulatory modalities in various disease states. Indeed, for both STN- and GPi-DBS our data suggests that improvement of tremor associates with electrode locations that are functionally connected to a common tremor response network, which is primarily centered around M1 and motor

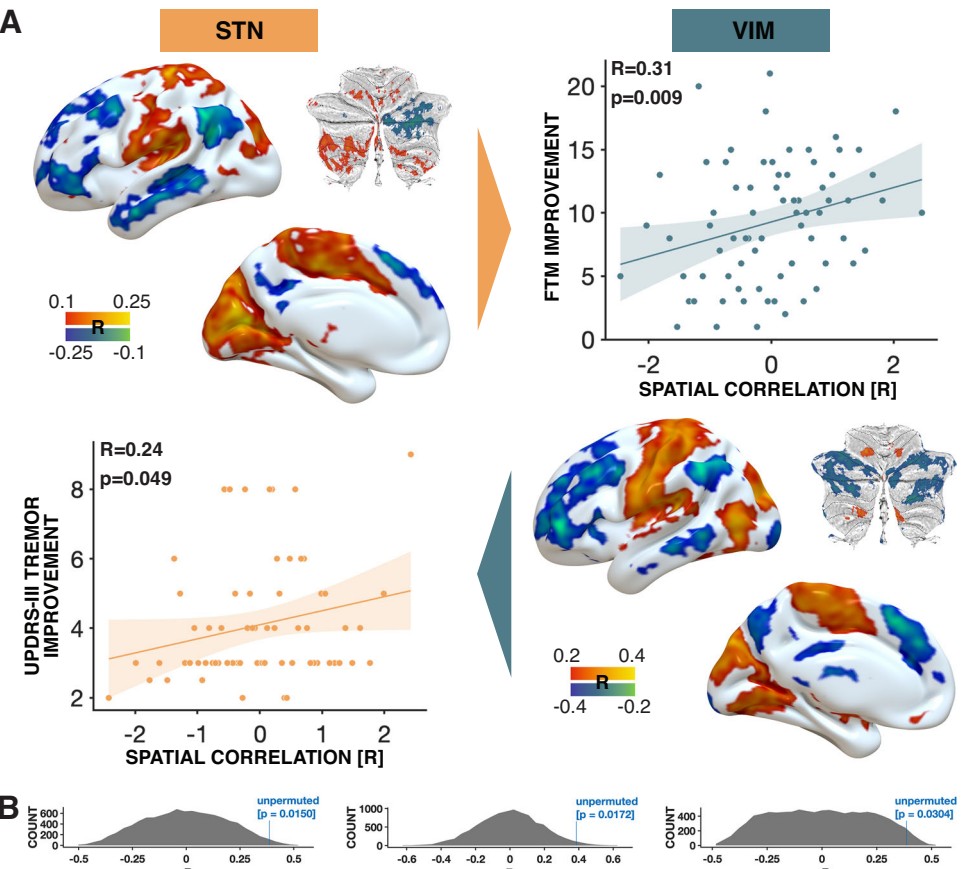

**Fig. 3 | Cross-validation and permutation analysis between DBS cohorts. A** Correlation maps for both the STN-DBS cohort (top left) and the VIM-DBS cohort (bottom right). These maps contain group-level correlation coefficients consisting of each patient's hemisphere connectivity profile with their clinical tremor improvement. The connectivity of VIM-DBS patients to the STN connectivity profile correlated positively with clinical improvement (top) in a two-sided Spearman correlation, and vice versa for STN patients with VIM profiles (bottom). No adjustment for multiple comparisons was applied. The shaded area indicates the 95% confidence interval. Cerebellar flatmaps were created using the SUIT toolbox[62]. **B** Results of a permutation analysis for VIM connectivity maps predicting STN-DBS outcomes (left), STN connectivity maps predicting VIM-DBS outcomes (right), and both combined (middle). DBS Deep Brain Stimulation, FTM Fahn-Tolosa-Marin Tremor Rating Scale, STN Subthalamic nucleus, UPDRS-III Unified Parkinson's Disease Rating Scale, part III, VIM Ventral intermediate nucleus of the thalamus.

cerebellum. Critically, the same tremor network resulted from *independently* analyzing PD and ET cohorts that underwent DBS, i.e., the shared topology of the network did not result from pooled data analysis.

Concurrent EMG-fMRI studies by Helmich et al. have also linked spontaneous fluctuations in PD tremor power with BOLD fluctuations in the cerebello-thalamo-cortical circuit[28,36,37]. According to this model, the basal ganglia would act as the "switch" for tremor onset and termination, while the cerebello-thalamo-cortical circuit functions as the "dimmer," modulating tremor amplitude. The interplay occurs via a transcortical pathway from the GPi through the ventral lateral anterior thalamus (VLa) to the motor cortex, which would then project to the cerebellum. Further, a bisynaptic subcortical pathway has been described that connects the STN via pontine nuclei to the cerebellum[38,39].

Our results indicate that targets outside the classic cerebello-thalamo-cortical circuitry, such as the GPi, may achieve tremor suppression by modulating network-level dynamics. This reinforces the concept that effective neuromodulation depends on the integration of multiple neural circuits rather than a single anatomically defined region. The principle of lesion network mapping provides a useful analogy, as it demonstrates that functionally connected regions, rather than isolated anatomical sites, drive symptom expression. In the case of Parkinsonian tremor, GPi and the basal ganglia circuitry is already seen as a 'switch' that may influence the cerebello-thalamo-

cortical circuit indirectly through polysynaptic pathways. We have devoted an extensive supplementary section including additional analyses and literature reviews on this important topic in section S1.

The present work supports the hypothesis that both rest and action tremors may unfold−likely in different patterns across time−along a shared anatomical network, even though abnormal activity in this common network may be triggered by different mechanisms in different diseases (as is apparent in different clinical phenotypes). A shared anatomical topography does not imply that the dysfunctional oscillations that unfold within the anatomically constrained networks are the same. From clinical observations, for instance, we know that rest tremor occurs at rest while action tremor occurs while carrying out actions. So, across the temporal domain, signatures of these two distinct symptoms will certainly be different−but may still unfold across a shared anatomical network. We pose this hypothesis since connectivity patterns associated with tremor improvements were similar across tremor types, for instance in VIM-DBS for ET and STN-DBS for PD. Additionally, repeating analyses for different forms of tremor within the out-of-sample cohort suggested a shared network might be involved onto which these tremors unfold. Moreover, this shared network may be associated with tremor improvements in different modalities, such as naturally occurring brain lesions and DBS. This convergence again aligns with our hypothesis of a common tremor network and underscores not only the critical role of functional connectivity between the basal ganglia and cerebello-thalamo-cortical

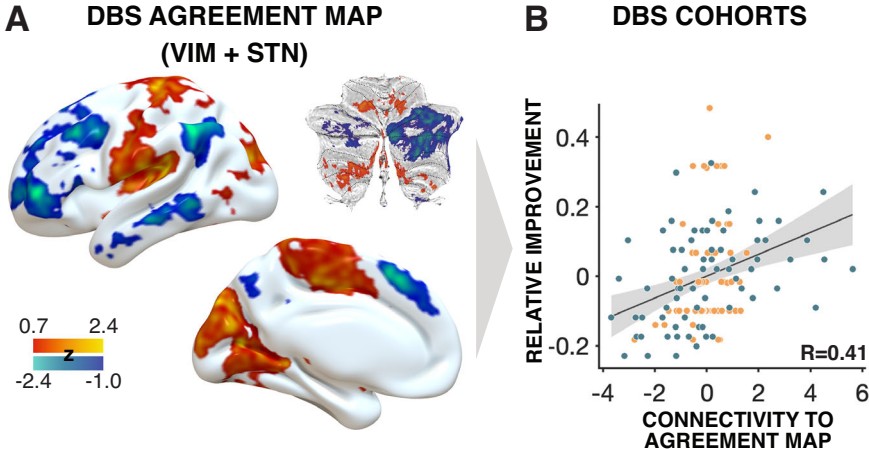

**Fig. 4 | *DBS Agreement map*. A** An *agreement map* was calculated by retaining connections that were positively or negatively associated with optimal clinical outcome in both the STN and the VIM cohorts, respectively. Absolute values of retained voxels were multiplied and standardized (z-scored). Cerebellar flatmaps were created using the SUIT toolbox[62]. **B** Spearman correlation (two-sided) between connectivity of DBS sites to the agreement map and clinical improvements (*R* = 0.41). This analysis was circular and meant to demonstrate the degree of fit between model and data, i.e., the maximal possible amount of variance that could be explained by this agreement map in the data used to create it. Due to circularity, we avoid reporting the *p*-value of this correlation. However, correlations remained significant when subjecting the analysis to fivefold cross-validation, which circumvented circularity (rho(135) = 0.20, *p* = 0.018, 95% CI [0.04, 0.36]). No adjustment for multiple comparisons was applied. The shaded area indicates the 95% confidence interval. DBS Deep Brain Stimulation, STN Subthalamic nucleus, VIM Ventral intermediate nucleus of the thalamus.

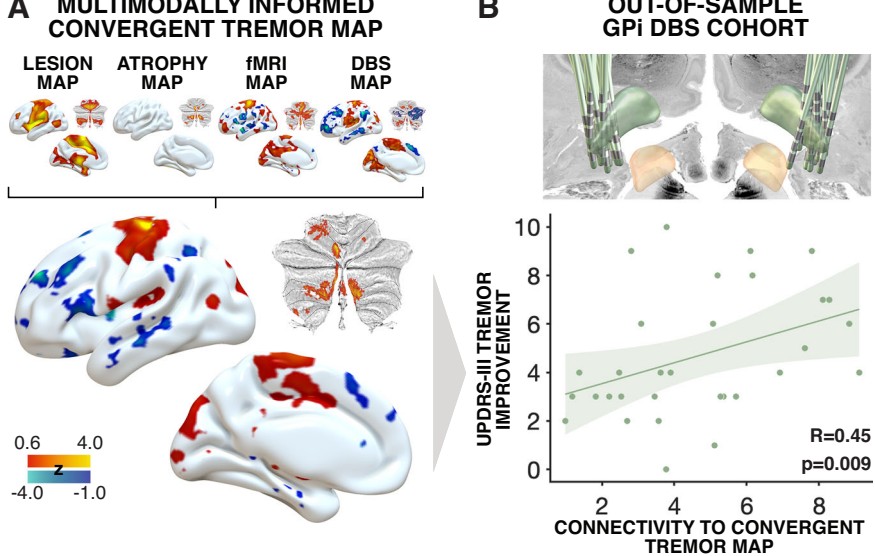

**Fig. 5 | Multimodally-informed convergent tremor map anticipates clinical outcomes in out-of-sample GPi cohort. A** The lesion-derived map, EMG-fMRI-derived map, VIM-STN (agreement) map and an ET-specific atrophy map were superimposed to create the displayed multimodally-informed convergent tremor map, with z-scores visualized. Cerebellar flatmaps were created using the SUIT toolbox[62]. **B** In an out-of-sample cohort of 31 analyzed hemispheres from PD patients with GPi-DBS, lead localizations were conducted using Lead-DBS (localizations shown in the top right image)[56]. Spearman correlation (two-sided) between connectivity of DBS sites to the convergent tremor map and MDS-UPDRS-III tremor improvements. No adjustment for multiple comparisons was applied. The shaded area indicates the 95% confidence interval. ET Essential tremor, DBS Deep Brain Stimulation, GPi Globus pallidus internus, STN Subthalamic nucleus, UPDRS-III Unified Parkinson's Disease Rating Scale, part III, PD Parkinson's disease, VIM Ventral intermediate nucleus of the thalamus.

circuits in effective tremor modulation, but also the potential for symptom-specific neuromodulation strategies. Targeting this shared network—via invasive or noninvasive forms of neuromodulation[40,41]—could enhance therapeutic outcomes across a range of tremor-related disorders, offering a treatment approach that is not dependent on the specific etiology of the tremor. A key cornerstone of this network should be seen in the cerebellothalamic pathway—or dentato(rubro) thalamic tract (DRT), for which extensive invasive neuromodulation evidence exists regarding its role in alleviation of tremor in ET, PD, and other conditions[6,20]. Indeed, an observational case series published by

Coenen et al. compiled evidence that this tract could serve as a DBS target for ET, PD tremor, dystonic head tremor, Multiple sclerosis tremor, as well as tremor in tardive dystonia and infarction[20]. Other reports have successfully targeted the tract or adjacent VIM for treatment of orthostatic tremor[42] or Holmes tremor[43]. As a central component of the here-described convergent tremor network, this fiber bundle connects cerebellar outflow tracts with the thalamus and is certainly a critical component in the treatment of tremor[6,7]. Consistently, we found that the active contacts of VIM-DBS electrodes from both the Berlin cohort and the Florida cohorts aligned with the

trajectory of the DRT, reinforcing its relevance as a key anatomical substrate for effective tremor control (see Fig. S3).

As a critical point for discussion, it should be emphasized that while we stick to the clinical convention of referring to VIM-DBS for the ET cohorts, the actual target region differs across centers and may not always focus exclusively on the ventral intermediate nucleus, proper[30]. Indeed, many centers choose to predominantly activate contacts below the thalamus, in an anatomically complex region that hosts both grey and white matter structures, and which has sometimes been descriptively termed the PSA[6,7]. This distinction can be seen across the two cohorts included here, where electrode placement in one centered around the VIM proper, while the second on the PSA (Figs. S4 and S5). Moreover, even the concept of the VIM as a nucleus has been questioned by some researchers, who considered it more of a transitional zone between afferent pallidal and cerebellar fibers rather than a distinct entity[44]. Complicating matters further, the region covered by the nucleus has received different names by different anatomists and not all segregated it in the same fashion[6]. Shedding light on this complex discussion goes beyond the scope of our article – for a recent review, see[6]. However, analyzing both VIM-DBS cohorts included here separately showed highly similar optimal connectivity profiles, despite their differing average stimulation coordinates (Figs. S4 and S5). Importantly, results remained overall consistent when including both cohorts in a supplementary replication of the main analysis (Fig. S6), reinforcing the robustness of the identified tremor network.

In general, functional networks derived from single disorders or modalities are limited in their applicability and often fail to translate across diseases or neuromodulation approaches. For instance, while DBS for ET may benefit from targeting a specific circuit, it is unlikely to inform transcranial electric stimulation strategies for PD. A multimodal approach as taken here, which integrated data from fMRI, atrophy maps, and DBS studies may provide a more robust network capable of addressing patient and disease heterogeneity and generalizing to unseen modalities or disorders. By embracing this heterogeneity, we aimed to ensure our findings will be broadly applicable to various tremor-related disorders and stimulation modalities. Importantly, the principles of mapping circuits using various causal sources of information (referred to as convergent causal mapping[25]) extend beyond tremor disorders. For example, in neuropsychiatric disorders, a treatment network overlap has been identified between obsessive-compulsive disorder and obsessive-compulsive behavior in Tourette's Syndrome[45,46]. Moreover, networks related with tic-reduction in secondary tic disorders and Tourette's Syndrome aligned, as well[47,48]. Networks associated with depression have been shown to be shared across Major Depressive Disorder, PD, epilepsy, multiple sclerosis and stroke[25,49]. These findings further validate the potential of symptom-specific brain networks to inform therapeutic strategies across diverse conditions, underscoring the utility of a multimodal approach to neuromodulation[50].

While our linear regression model accounted for cohort differences and their interaction with connectivity, it is important to note that the model explained only a modest portion of the variance in clinical improvement (adjusted $R^2 = 0.13$ for the lesion-derived map and $R^2 = 0.13$ for the EMG-fMRI derived map). This suggests that other unmeasured factors, such as individual patient characteristics, differences in disease pathology, or variability in clinical scales, may play a significant role. We approximate the impact of these factors according to literature reports in Fig. S1. Additionally, the variability in clinical scales (e.g., MDS-UPDRS-III for PD vs. FTM-Tremor scale for ET) may influence the sensitivity of our analysis and should be considered when interpreting these results. Importantly, most factors influencing our findings are intrinsic or fixed and therefore not modifiable through clinical practice. The only exceptions are electrode placement and stimulation parameters, which directly shape the defined tremor response target. As such, this target serves as a

reference point with the potential to guide and enhance patient care[26].

Several limitations should be considered when interpreting our findings. First, the use of data from different patient populations for lesion network mapping and DBS analyses introduced variability. Although we focused on identifying commonalities across modalities, differences in disease progression, specific symptom components, treatment history, and patient characteristics may impact the generalizability of our results. Furthermore, the inherent variability in clinical scales across cohorts may also limit the interpretation of findings. However, as discussed, this heterogeneity was deliberately chosen to increase robustness of findings. As outlined in previous publications, any network conclusion that is primarily based on a single dataset of a single nature (e.g., STN-DBS alone) would be inherently biased towards this type of data[21]. Previous studies in different symptoms could illustrate that the convergent analysis of heterogeneous datasets typically improve robustness and generalizability of findings[24,25,48,50]. Our study incorporated diverse datasets across multiple DBS targets and integrated multimodal approaches, enabling a comprehensive synthesis of findings in tremor. While we deliberately embraced heterogeneity to enhance the robustness of our findings, we also applied inclusion criteria, such as requiring a tremor score above two, to reduce noise in the data. Additionally, all cohorts were also analyzed separately, and separate analyses were conducted to ensure that target-specific and shared networks were accurately identified.

Second, our study primarily relies on correlational data and is not inherently designed to be predictive. However, the occurrence of lesions or the onset of DBS and the subsequent symptom changes suggest a stronger causal relationship than mere correlation. Integrating findings from lesions, DBS, and fMRI within a single study could significantly strengthen causal inferences[21]. For instance, if lesioning a specific brain region disrupts a particular function and stimulating the same region with DBS produces a similar outcome, this convergent evidence would robustly support a causal link between that brain region and the observed function. Additionally, EMG-fMRI data may complement these findings by revealing the broader network-level effects of lesions and stimulation, thus providing deeper insights into the circuit mechanisms underlying the behavioral changes.

Third, while fMRI data offers valuable insights, its relatively slow temporal resolution does not capture the rapid dynamics of neural activity associated with tremor. Nonetheless, our findings agree with electrophysiological data, which offer complementary evidence of the neural mechanisms at play[34,35]. Indeed, they align with EEG and MEG approaches that focus on the cycle-by-cycle 4–8 Hz rhythmic activity, although these methods are hampered by a lower spatial resolution in subcortical brain areas relevant to tremor, such as the VIM[37,51]. Integrating connectomic analysis with these electrophysiological findings and prior research may provide a more comprehensive understanding of tremor networks, in the future.

In summary, the present study may advance our understanding of the brain network underlying tremor genesis and suppression by identifying a convergent tremor network and highlighting the potential for symptom-specific (rather than disease-specific) neuromodulation[47]. Prospective studies are needed to validate the utility of this network for potential translational applications.

## Methods
### Patients and cohorts
This study was conducted in accordance with the Declaration of Helsinki and was approved by the institutional review board of the Brigham and Women's Hospital, Boston, Massachusetts. The network mapping analysis was exempted from obtaining informed consent based on the secondary use of previously published data (Master vote

2020P002987). Use of patient data in the original studies was approved by the respective Institutional Review Boards. Ethical approval for the Berlin and Würzburg cohorts was initially granted by the Institutional Review Board of Charité−Universitätsmedizin Berlin (EA2/186/18). The Amsterdam cohort was recruited under approval from the Ethics Committee of Amsterdam Medical Center (MEC 06/084 # 07.17.0069), and the Shanghai cohort received approval from the Ethics Committee of Ruijin Hospital (2018017). Patients with PD and ET who had undergone DBS were retrospectively enrolled. Data stemmed from previously published PD-DBS cohorts from DBS centers in Berlin, Würzburg, Amsterdam, and Shanghai[22,52,53]. The ET cohort consisted of patients operated in Berlin[23]. As tremor may often occur as an asymmetric symptom, we included the tremor scores as hemiscores and assigned them to the contralateral DBS electrode, respectively, as done previously[12,23]. Data from patients with less than 3 tremor points were excluded as done previously[26], since substantial baseline symptoms are required to measure improvement by DBS, properly. To measure tremor severity and its changes, we employed validated scales for ET and PD. A second ET cohort of patients that underwent VIM-DBS at Jacksonville, Florida[12] served as a validation cohort. For ET, we used the Fahn-Tolosa-Marin (FTM[54]) Tremor Rating Scale, while for PD, we used the MDS-UPDRS, part III[55]. Subscales assessed rest, postural, and action tremor on a 5-point scale (0–4) for each hand. In the Berlin, and Amsterdam PD cohorts, the original UPDRS-III version was used, providing a combined score for action and postural tremor. Also, the constancy of rest tremor was not consistently available across cohorts. For a final hold-out test sample cohort we included patients with DBS to the globus pallidus internus (GPi). These data consisted of cohorts operated at DBS centers in Amsterdam and Shanghai. Supplementary Table S1 summarizes all patient cohorts.

## Deep brain stimulation electrode localization and calculation of stimulation volumes

Electrodes in the three cohorts were localized using default settings in Lead-DBS software version 3[56]. Briefly, preoperative MRI and postoperative CT or MRI scans were linearly co-registered using advanced normalization tools (ANTs, http://stnava.github.io/ANTs/). To minimize bias introduced by a non-linear deformation of the brain due to pneumocephalus, the brain shift correction step in Lead-DBS was carried out. Multi-spectral preoperative volumes were then used to compute a spatial normalization warp field into ICBM 2009b Nonlinear asymmetric ('MNI') space using the SyN Diffeomorphic Mapping approach implemented in ANTs. Normalization results were checked and in cases where normalization errors were clearly visible and image quality allowed refinement, the integrated toolbox WarpDrive was used to reach optimal normalization results for subcortical structures[57]. Subsequently, DBS electrodes were localized using the PaCER algorithm for CT volumes ($N = 97$) or the TRAC/CORE method for MRI volumes ($N = 27$). Results were carefully inspected and manually refined, if necessary, using Lead-DBS. Anatomical segmentations of subcortical structures shown in the present manuscript were defined by the DISTAL atlas using the Lead Group analysis tool[58]. Estimations of VTA around electrodes were calculated applying a finite element method-based model in each patient[22].

## A priori published network maps

Three published tremor network maps were used to predict ranks of DBS improvements in aforementioned DBS cases. First, a lesion network map published by Joutsa et al.[19] was used which had been calculated by seeding normative functional connectivity from lesions that alleviated tremor in patients with ET. This map includes values for each voxel, representing the overlap of voxels functionally connected to all 11 lesion locations. A higher number of lesions connected to a specific voxel corresponds to a higher probability of tremor alleviation for the

specific voxel. Second, a conjunction network map derived from atrophy network mapping in ET patients was used[27]. This map included a network hub that was both sensitive and specific to ET. Third, the map derived from concurrent EMG-fMRI scans was based on PD patients and led to the 'dimmer-switch' model of tremor[28]. A detailed description of the methods used to generate the EMG-fMRI map is provided in the supplementary material (see supplementary methods S1).

## Statistical analyses

**Connectome calculation.** For the purpose of this study, a normative connectome was created based on time series of resting state fMRI acquired in 1087 healthy subjects from the Human Connectome Project (http://www.humanconnectomeproject.org) cohort. The data had been acquired using specialized magnetic resonance hardware[59] and are publicly available in 'minimally preprocessed' form in MNI space. Time series in four runs per subject acquired at a repetition rate of 720 ms were concatenated after mean-averaging each run, then correlation coefficients between each pair of 2 mm isotropic voxels in MNI space were calculated, leading to a 285,903 × 285,903 adjacency matrix per subject. Matrices were averaged across the 1087 subjects to embody a normative 'group connectome' that was used to estimate connectivity profiles seeding from each DBS stimulation field, going forward. To evaluate the replicability of our findings, we repeated the analysis using a disease-specific connectome derived from MRI data of 90 patients from the Parkinson's Progression Markers Initiative database[22,60].

**Deep brain stimulation network mapping and validation.** The functional connectivity profile ('seedmap') of each VTA was calculated using the Lead Connectome Mapper tool included within Lead-DBS[22] after non-linearly flipping right-sided VTA to the left side. When using a preprocessed connectome in 'matrix' format (see above), this tool averages rows of averaged connectivity coefficients that fall into the domain of the seed region (VTA), which results in a single value per MNI voxel and hence denotes average connectivity between the seed region and the rest of the brain. Resulting connectivity profile maps ('seedmaps') for each electrode's VTA, thus each hemisphere, therefore included (average) positive and negative functional connectivity values between the VTA and other voxels in the brain. Each electrode's *seedmap* was associated with the corresponding (contralateral) improvement score.

**Estimating DBS outcomes based on published a-priori tremor maps.** Once *seedmaps* from VTAs were created, they could be compared with other maps by means of summing up voxel-wise multiplications. This metric measures how strongly two seeds are connected with one another. Using this method, first, *seedmaps* were compared with the two a priori maps described above, i.e., the lesion network map[19] and the EMG-fMRI map[37]. This process led to a coefficient for each electrode for both of the two a-priori maps. Coefficients were correlated with clinical hemiscore improvements to test whether it was associated with optimal response if VTAs were connected to a priori maps.

**Estimating DBS outcomes across DBS targets and indications.** In a next step, we created data-driven maps of optimal connectivity based on either of the VIM- or STN-DBS cohorts. This was done following the approach in Horn et al.[22], i.e., by correlating connectivity values in the seedmaps with hemiscore improvements across the cohort of patients. This resulted in correlation maps (*R-maps*) of 'optimal' connectivity for either of the two cohorts. These *R-maps* denoted positive correlation coefficients for regions that were positively connected predominantly to electrodes of top responders and negative coefficients for the ones

predominantly connected to electrodes in poor responding patients. To validate each *R-map*, a fivefold cross-validation was applied within each cohort. This means that patients were randomly assigned to fivefolds and, iteratively, the *R-map* was recalculated after leaving out one fold. Then, similarities between *seedmaps* of the left-out fold and the respective *R-map* were calculated using voxel-wise spatial correlations and subsequently correlated with clinical hemiscore improvements[22]. To explore the differences and similarities of optimal network profiles in STN- and VIM-DBS, we next tested whether one map could predict ranks in outcomes of the respective other cohort. Again, each *seedmap* was spatially correlated with the *R-map* of the respective other cohort and coefficients were subsequently correlated with clinical outcomes. To address the potential influence of varying surgical approaches, an optimal connectivity profile (*R-map*) was calculated using an additional ET cohort from Jacksonville, Florida. This cohort was compared to the ET cohort from Berlin to assess consistency. Importantly, the Florida cohort was not included in the primary analyses and served exclusively as an independent validation check.

**Integrating results into a multimodal convergent tremor map.** Up until here, five maps had been analyzed: two data driven maps from the two DBS cohorts, as well as the three a priori maps (lesion network mapping, atrophy network mapping, and EMG-fMRI). All five maps looked highly similar by visual inspection, with the exception that the atrophy map only defined cerebellar regions (which showed high overlaps with the other maps). To identify regions consistently associated with tremor alleviation across the five different maps, regardless of target or modality, we calculated an *'agreement map'* following the concept by Sobesky et al.[24]. Namely, this simple approach retains voxels that are either positive or negative across all maps, while discarding other areas[24]. To preserve weighting, the retained values are multiplied across maps while maintaining the sign. This process was first done across the two DBS maps (STN-DBS and VIM-DBS), and later across both maps and the three a priori maps, resulting in a multimodally informed convergent tremor map. In this process, the atrophy network map only informed cerebellar voxels since it did not include any cerebral voxels. Voxels that were preserved in the resulting maps were positively or negatively associated with tremor across all included maps.

**General statistical assumptions.** For correlation analyses, Spearman's rank correlation coefficients were calculated throughout, and the *p*-value was calculated based on permutation testing ($\times 5000$). To make scores comparable across PD and ET, analyses that included both cohorts were performed after normalizing absolute improvements by the maximum reachable tremor score (which differs in the UPDRS-III and FTM). All analyses controlled for study cohorts using a dummy-regressor. A graphical overview of DBS network mapping methods is provided in supplementary Fig. S11.

**Reporting summary**
Further information on research design is available in the Nature Portfolio Reporting Summary linked to this article.

## Data availability
Due to privacy regulations concerning patient health information, patient-specific imaging data is not publicly available but can be obtained from the corresponding author upon request. The connectivity data generated in this study are provided in the Source Data file. Atlases used for visualization are openly available within Lead-DBS software (www.lead-dbs.org). Source data are provided with this paper.

## Code availability
All code that was used in the present study is openly available within Lead-DBS software (https://github.com/netstim/leaddbs).

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

## Acknowledgements

Data were provided in part by the Human Connectome Project, WU-Minn Consortium (Principal Investigators: David Van Essen and Kamil Ugurbil; 1U54MH091657) funded by the 16 NIH Institutes and Centers that support the NIH Blueprint for Neuroscience Research; and by the McDonnell Center for Systems Neuroscience at Washington University. The study was funded by Deutsche Forschungsgemeinschaft (DFG, German Research Foundation)—Emmy Noether Stipend 410169619, Project-ID 424778381—TRR 295). L.L.G. was participant in the BIH Charité Junior Clinician Scientist Program funded by the Charité - Universitätsmedizin Berlin and the Berlin Institute of Health at Charité (BIH). L.L.G. and B.H.B. were supported by the Thiemann foundation. P.Z. received support from a scholarship from the Einstein Center for Neurosciences Berlin. J.J. received grants from the Research Council of Finland, Sigrid Juselius Foundation, Finnish Medical Foundation, Sakari Sohlberg's Foundation, Finnish Foundation for Alcohol Studies, Ane and Signe Gyllenberg

Foundation, Finnish Parkinson Foundation, University of Turku (Sigrid Juselius Foundation, Private Donation) and Turku University Hospital (VTR funds); M.D.F. was supported by grants from the NIH (R01MH113929, R21MH126271, R56AG069086, R21NS123813, R01NS127892, R01MH130666, UM1NS132358), Neuronetics, the Kaye Family Research Endowment, the Ellison/Baszucki Family Foundation, and the Manley Family. A.H. was supported by the Schilling Foundation, the German Research Foundation (Deutsche Forschungsgemeinschaft, 424778381—TRR 295), Deutsches Zentrum für Luft- und Raumfahrt (DynaSti grant within the EU Joint Programme Neurodegenerative Disease Research, JPND), the National Institutes of Health (R01MH130666, 1R01NS127892-01, 2R01 MH113929 & UM1NS132358) as well as the New Venture Fund (FFOR Seed Grant).

## Author contributions

L.L.G., M.D.F., and A.H. conceptualized the study. L.L.G., B.A., N.L., L.S., C.N., and B.B. performed electrode localization and data analysis. N.L. and A.H. calculated the connectome. P.Z., M.R., J.V., C.Z., E.H.M., V.J.J.O., R.M.A.B., E.F.P.Y., D.T.C., J.J., M,D, R.C.H., and A.A.K. analyzed clinical data from the different cohorts that contributed to the lesion-, atrophy-, fMRI-, and DBS- map. R.C.H., G.D., A.A.K., M.D.F., and A.H. discussed the results and research direction. L.L.G. wrote the manuscript with input and revisions from all authors.

## Competing interests

A.H. reports lecture fees for Boston Scientific, is a consultant for Modulight.bio, was a consultant for FxNeuromodulation and Abbott in recent years and serves as a co-inventor on a patent granted to Charité University Medicine Berlin that covers multisymptom DBS fiberfiltering and an automated DBS parameter suggestion algorithm (patent #LU103178) unrelated to present work. M.D.F. has intellectual property on the use of brain connectivity imaging to analyze lesions and guide brain stimulation; is a consultant for Magnus Medical, Soterix, Abbott, and Boston Scientific; and has received research funding from Neuronetics unrelated to present work. J.J. received congress travel support from Insightec, Abbott, and AbbVie; lecturer honoraria from Addiktum, Insightec, Nordic Infucare, Lundbeck, and Novartis; and consultancy fees from Adamant Health, Summaryx, and TEVA Finland; and owns stock of Neurologic Finland and Suomen Neurolaboratorio and acts as an advisory board member for TEVA Finland unrelated to present work. A.A.K. reports lecturer honoraria and consultancies from Boston Scientific and Medtronic unrelated to present work. The remaining authors declare no competing interests.

## Additional information

[1]Center for Brain Circuit Therapeutics, Department of Neurology, Brigham & Women's Hospital, Harvard Medical School, Boston, MA, USA. [2]Department of Neurology with Experimental Neurology, Charité – Universitätsmedizin Berlin, Corporate Member of Freie Universität Berlin and Humboldt- Universität zu Berlin, Berlin, Germany. [3]Department of Pediatric Neurology, Charité – Universitätsmedizin Berlin, Corporate Member of Freie Universität Berlin and Humboldt- Universität zu Berlin, Berlin, Germany. [4]Department of Neurology, Center for Movement Disorders and Neuromodulation, Medical Faculty and University Hospital Düsseldorf, Heinrich Heine University Düsseldorf, Düsseldorf, Germany. [5]Institute of Clinical Neuroscience and Medical Psychology, Medical Faculty and University Hospital Düsseldorf, Heinrich Heine University Düsseldorf, Düsseldorf, Germany. [6]Einstein Center for Neurosciences Berlin, Charité – Universitätsmedizin Berlin, Corporate Member of Freie Universität Berlin and Humboldt- Universität zu Berlin, Berlin, Germany. [7]Department of Neurology, University Hospital Würzburg, Würzburg, Germany. [8]Department of Neurosurgery, Ruijin Hospital affiliated to Shanghai Jiao Tong University School of Medicine, Shanghai, China. [9]Department of Neurology, Amsterdam University Medical Centers, University of Amsterdam, Amsterdam, The Netherlands. [10]Cognitive Neuroscience Unit, School of Psychology, Deakin University, Geelong, VIC, Australia. [11]Turku Brain and Mind Center, Clinical Neurosciences, University of Turku, Turku, Finland. [12]Department of Radiology, Mayo Clinic, Jacksonville, FL, USA. [13]Neurocenter, Turku University Hospital, Turku, Finland. [14]Department of Neurology, Center of Expertise for Parkinson and Movement Disorders, Donders Institute for Brain, Cognition and Behaviour, Radboud University Medical Centre, Nijmegen, The Netherlands. [15]Department of Neurology, UKSH, Christian-Albrechts-University Kiel, Kiel, Germany. [16]MGH Neurosurgery & Center for Neurotechnology and Neurorecovery (CNTR) at MGH Neurology Massachusetts General Hospital, Harvard Medical School, Boston, MA, USA. [17]Network Stimulation Institute, Department of Stereotactic and Functional Neurosurgery, University Hospital Cologne, Cologne, Germany. ✉e-mail: goede.lukas@gmail.com

