## [Transparent Peer Review file · Nature Communications]

Convergent mapping of a tremor treatment network

Corresponding Author: Dr Lukas Goede

Version 0:

Reviewer comments:

Reviewer #1

(Remarks to the Author)

Manuscript Title: Convergent causal mapping of a tremor treatment network

1. There is an extra comma between “neuromodulation” and “in the future” on Line 61.

2. There is an extra comma between “mapping” and “studies” on Line 94.

3. If DBS network maps from previous studies across various targets and disorders already indicate tremor benefits, can the authors elaborate on the additional clinical advantages of producing a “multimodal tremor network” that combines data from functional MRI, atrophy, and DBS studies?

4. In the results section, the authors report that the degree of spatial similarity between a stroke-lesion-based tremor-relief map and a connectivity map derived from VTA correlates with relative improvement.

5. For the above point (Figure 1), could the authors provide an example of:

(a) A VTA-derived connectivity map (seed map) that exhibits high spatial similarity with the stroke-based lesion map.

(b) A map with low spatial similarity as a reference.

6. It appears that the stroke-lesion-based tremor-relief map represents significance (p-value) of tremor relief, while the VTA-derived seed map depicts average connectivity coefficients (r-value). Given that these are distinct statistical metrics, how was spatial similarity computed between the two maps?

7. In Figure 1B and Figure 2B, the observed trend appears relatively weak ($r = 0.25$ for both figures), though supported by a larger sample size through the combination of data from VIM and STN DBS targets. Do similar results hold when the analysis is performed separately for VIM and STN? Alternatively, would a multivariate regression analysis with STN/VIM grouping as a covariate provide deeper insights?

8. As noted by the authors, habituation is reported in VIM-DBS for ET patients, which might otherwise diminish tremor benefits at longer postoperative time points. Did the authors account for the time elapsed since surgery in analyses correlating postoperative tremor benefits with connectivity maps?

9. In disorders like Parkinson’s Disease (PD) or Essential Tremor (ET), numerous studies have demonstrated differences in fMRI/BOLD signals compared to healthy cohorts. Given that this analysis utilized group-level resting-state fMRI data from a healthy subject group, how did the authors account for disease-specific changes in BOLD activity independent of stimulation effects? Are there fMRI-based connectomes specific to ET and PD cohorts that could be used to replicate the findings?

10. In the discussion, the authors state that “both rest and action tremors may unfold within the same anatomical network.” To better support this claim, the authors might consider using the “convergent tremor map” and quantifying connectivity to this map. This could then be correlated with not only total UPDRS-III tremor improvement but also with its action, postural, and rest tremor subscores.

Reviewer #2

(Remarks to the Author)

The manuscript establishes a foundation by addressing the diversity of tremor-related disorders, such as Parkinson’s Disease (PD) and Essential Tremor (ET), characterized by rhythmic, involuntary oscillations. While these disorders have distinct origins, the authors hypothesize a shared network underpinning tremor manifestations, a premise this study seeks to investigate. Traditional therapeutic approaches have targeted specific brain regions (e.g., the ventrointermediate nucleus (VIM) and the posterior subthalamic area (PSA)). However, emerging research proposes that tremor may arise from pathological activity within a distributed network rather than isolated locations. The authors aim to map a convergent network for tremor treatment by integrating lesion mapping, atrophy patterns, task-based fMRI data, and deep brain stimulation (DBS)

network data. This ambitious approach seeks to uncover a unified treatment framework spanning various tremor types and DBS targets.

1. Combining PD and ET Populations: While the network-based approach is commendable, does it sufficiently justify merging PD and ET populations with potentially distinct etiologies and network origins? Specifically, as the results and title suggest a causative network, is it appropriate to generalize across these tremor types?
2. Etiology-Specific Networks: Shouldn't the study seek to identify distinct networks for tremor based on etiology? While rhythmic oscillatory patterns may reflect similar phenomenology, PD tremor typically occurs at rest, whereas ET tremor manifests during voluntary movement. Could analyzing these disorders together obscure critical distinctions and introduce confounding variables?
3. Hypotheses on Network Convergence: The introduction alludes to shared tremor-relief networks across conditions. How do the authors specifically hypothesize these networks will converge or diverge between disorders?
4. Details of Task-Based fMRI: The manuscript references task-fMRI maps but does not specify the motor tasks involved. This is critical because nearly all motor tasks engage the cerebello-thalamo-cortical circuit, widely recognized as the final common pathway for movement. Are the tasks designed to elicit tremor-related BOLD activity? Details on rest vs. action tremor paradigms, specific motor events, and the participant populations (PD and ET patients or healthy controls) should be explicitly clarified.
5. fMRI method: Would fMRI be considered a good method for specifying network activity related to an oscillatory phenomenon once BOLD effect is an indirect method to show brain regional activation and also features low temporal resolution?

The study includes PD and ET patients undergoing DBS targeting the STN, VIM, and GPi across multiple international centers. Lesion maps, atrophy patterns, and task-fMRI maps were used as reference networks to assess connectivity across these DBS targets. DBS electrode locations and volumes of tissue activated were modeled using the Lead-DBS software. Data-driven maps derived from each cohort were cross-validated and integrated to create a unified, multimodal "agreement map." Statistical analyses primarily relied on Spearman's rank correlations and permutation tests.

1. Sample Size Adequacy: Are the sample sizes for each tremor population sufficient to support the observed correlations and ensure generalizability?
2. Merging Data Across Disorders: The integration of PD and ET datasets for network analysis assumes shared pathophysiology. However, combining these populations risks oversimplifying distinct disease mechanisms. Would separate analyses yield more nuanced insights?
3. Normative vs. Individual Connectivity Variability: Given that Lead-DBS and its associated connectome rely on normative data, could individual patient variability in connectivity affect reproducibility?

The results indicate varying levels of tremor improvement across DBS targets, with STN-DBS (PD) showing 87.4% improvement, VIM-DBS (ET) 67.0%, and GPi-DBS (PD) 90.1%. Significant associations were observed between tremor improvement and connectivity to lesion-based tremor relief maps and fMRI tremor maps, but not with atrophy maps. Cross-validation between STN and VIM cohorts revealed predictive capabilities, suggesting an overlapping network. However, connectivity-to-outcome correlation coefficients were relatively low ($R=0.11$ to $R=0.45$).

1. Predictive Power of Connectivity Studies: Low correlation coefficients could indicate weak predictive power for connectivity studies in forecasting tremor improvement. Would refining these models improve their reliability?
2. Separate Analyses for DBS Targets: Would distinct analyses of STN, VIM, and GPi targets, as well as separate evaluations of tremor types, provide deeper insights into DBS effectiveness?
3. Atrophy Maps and Predictive Limitations: The absence of significant associations with atrophy network maps highlights potential limitations in using this modality for tremor prediction. Could excluding these maps enhance the validity of the agreement map?
4. Lesion Sites vs. Atrophy Maps: The findings suggest lesion sites are more predictive than atrophy maps. How does this reconcile with the hypothesis of a network origin? Furthermore, GPi targeting for PD tremor, located outside the cerebello-thalamo-cortical circuitry, produced better outcomes than VIM targeting within the hypothesized network. Do these findings fully support the proposed conclusions?

The authors conclude that integrating lesion mapping, atrophy patterns, task-fMRI, and DBS data supports the existence of a symptom-specific tremor network, potentially enabling treatment independent of tremor etiology. This network reportedly involves the motor cortex, cerebellum, and thalamus, suggesting a pathway for symptom-specific DBS interventions. However, they acknowledge limitations stemming from the study's correlational design, heterogeneous populations, and multicenter data, which could introduce variability yet potentially enhance robustness.

1. Balancing Variability and Robustness: The study emphasizes the robustness achieved through diverse populations. Could stricter inclusion criteria or subgroup analyses mitigate variability and strengthen the network's validity?
2. Clinical Implications of Low Effect Sizes: Given the low effect sizes observed, what are the practical implications of targeting specific network hubs? Would a more focused analysis better clarify optimal DBS targets?
3. Role of Atrophy Maps: With atrophy maps showing no significant correlations, how does this influence the predictive value of the multimodal approach? Should their exclusion or refinement be considered for future research?

Final Concerns:

Premise and Hypothesis Misalignment

The central hypothesis of the manuscript suggests a "common and causative tremor network" rooted in the cerebello-thalamo-cortical circuit. However, this circuit is already widely acknowledged as the common final pathway for virtually all forms of movement, including tremor. The study does not present novel evidence to distinguish this network as uniquely causative for tremor over other motor phenomena.

Furthermore, while the cerebello-thalamo-cortical circuit is a logical focus for therapeutic targets such as Vim in DBS for tremor, the clinical results paradoxically demonstrate superior outcomes for STN and GPi DBS. These targets lie outside the cerebello-thalamo-cortical network and instead address pathological changes in the basal ganglia circuitry, particularly

those associated with dopamine deficiency in Parkinson's disease (PD). This discrepancy undermines the conclusion that the cerebello-thalamo-cortical network is the optimal therapeutic focus for tremor relief.

Implications of the Conclusions

The conclusions, if left unrefined, may lead readers to infer that DBS targets within the cerebello-thalamo-cortical circuit (e.g., Vim) are inherently superior or more mechanistically appropriate for tremor treatment. However, the observed efficacy of STN and GPi DBS in PD tremor challenges this narrative. Future research would benefit from focusing on DBS targets identified through mechanistic insights—such as addressing basal ganglia dysfunction in PD—rather than defaulting to the cerebello-thalamo-cortical network, which serves as a generic movement-related pathway.

Additional Limitations

Justification for Combining PD and ET Populations

The manuscript combines data from PD and essential tremor (ET) populations, which are known to have distinct etiologies and pathophysiological mechanisms. While the authors adopt a network approach, they provide insufficient justification for merging these cohorts, potentially oversimplifying divergent disease mechanisms. Separate analyses may offer a more nuanced understanding of tremor networks across these disorders.

Clarification of Clinical Relevance

The reported connectivity-outcome correlations are low to moderate in strength, raising concerns about their clinical significance. The authors should address whether these correlations meaningfully predict therapeutic outcomes and whether the modest effect sizes might limit the practical utility of their findings.

Lack of Significant Findings from Atrophy Maps

The study notes the absence of significant associations between tremor outcomes and atrophy maps. This raises questions about the role of structural degeneration in tremor pathophysiology and whether its exclusion would impact the multimodal network proposed. An explanation of these results, as well as their implications for the study's conclusions, is needed.

Reviewer #3

(Remarks to the Author)

In their work, Goede et al. aim to explore whether a network of brain regions is associated with the multifaceted manifestations of tremor. By integrating results from lesion-based studies, functional imaging, and research involving patients with implanted DBS leads, the authors seek to create a unified representation of a tremor network. While the study addresses an important and underexplored area, several aspects warrant further discussion or clarification.

The reported findings of a network involving the primary motor area (M1), the supplementary motor area (SMA), and the cerebellum are unsurprising and largely reiterate what is already well documented in the existing literature. However, particularly interesting is the demonstrated opportunity to compare the two cohorts using functional lesion-based methods at different sites. At this point, the group might clarify why less than 10% of the variance (R^2 , assuming this is a correlation coefficient) is explained, despite a visually strong agreement. This aspect was rather unexpected.

I believe the findings could be better contextualised if the clinical data were described more clearly. The tremor scores do not seem widely recognised. What was the rationale for not summing all tremor-related items of the UPDRS (MDS-UPDRS-III, Items 3.15-3.18) and just using action and rest tremor scores? According to what criteria were the leads implanted and the targets chosen—particularly considering that the ventrolateral thalamus is highly heterogeneous and, in my experience, surgical approaches vary? Were there also cases where the region below the posterior ventrolateral thalamus was targeted in ET patients? The close anatomical relationship between the posterior subthalamic areas—possibly representing the dentatorubrothalamic tract (DRT)—and the STN in PD patients, as well as the potentially high energy levels possibly even in upper (dorsal) contacts, could lead to poor differentiation between the networks and blur the results by Goede et al.

The use of thalamic nomenclature is another point of contention. The term "ventral intermediate nucleus (VIM)" has been questioned by some researchers, who consider it more of a transitional zone between afferent pallidal and cerebellar fibres rather than a distinct entity (for reference, see: <https://movementdisorders.onlinelibrary.wiley.com/doi/10.1002/mds.10136>). It is a bit surprising that the authors—some of whom have significantly contributed to prior work in this area—chose to adopt a less precise terminology I have never seen before, such as "ventrointermediate nucleus". Opting for an alternative nomenclature might provide a clearer representation of the anatomical structures under investigation.

Version 1:

Reviewer comments:

Reviewer #1

(Remarks to the Author)

Appreciate the very thorough and comprehensive responses

Reviewer #2

(Remarks to the Author)

I have carefully reviewed the authors' responses and acknowledge that most concerns have been adequately addressed. However, one critical issue remains unresolved, which could challenge the premise that a common tremor network represents a superior therapeutic target compared to those based on pathophysiologic circuitry, such as basal ganglia targets for Parkinson's disease (PD) or rest tremor.

Specifically, an important question from the initial review remains unanswered: How does GPi targeting for PD tremor—despite being outside the cerebello-thalamo-cortical circuitry—yield better outcomes than VIM targeting (data presented in the manuscript), which is located within the hypothesized tremor network? Do these findings fully support the proposed conclusions?

Reviewer #3

(Remarks to the Author)

Goede and his co-authors put an impressive amount of work into their revision of the manuscript now titled 'Convergent Mapping of a Tremor Treatment Network' and attempted to address the numerous questions raised by the three reviewers. Of particular note is the additional cohort treated at the Mayo Clinic Florida. One inevitably wonders why this cohort was used 'only' for validation and why the primary analyses were not supplemented. Looking at the active contacts (Figures S3/S4), they are very different. This raises the question of whether this difference affects the credibility of the results, as it seems almost irrelevant at what point the THS impulses arrive to activate the networks. Given the small effects observed, this is, of course, an important limitation to the overall results.

Version 2:

Reviewer comments:

Reviewer #2

(Remarks to the Author)

I would like to once again commend you on this outstanding work—one that very few research groups could currently produce with such depth and rigor. The organization, data collection from multiple centers, integration of lesion studies, multimodal data processing, and comprehensive discussion make this study an exceptional contribution to the field. The integration of diverse data sources (MRI, connectomics, lesion studies) not only strengthens hypotheses related to DBS targeting but also offers crucial insights into circuit-level mechanisms underlying neurological symptoms.

Your work convincingly demonstrates, through robust data, that tremor across different etiologies is fundamentally linked to abnormal oscillatory activity within the cerebello-thalamic circuit, which appears essential for its manifestation. The manuscript effectively illustrates how different tremor types converge on a shared neural network, regardless of whether their initial pathological origin lies within adjacent circuits such as the basal ganglia (e.g., Parkinsonian tremor).

However, while the study establishes a strong case for a convergent tremor network, it is less conclusive in defining it as a universal therapeutic target. The success of GPi DBS—despite targeting a structure outside the cerebello-thalamic circuit—suggests that tremor treatment does not necessarily require direct modulation of this network. Instead, pathological activity may originate externally but propagate through the cerebello-thalamic loop, where it manifests as tremor. Similarly, this may be the case for STN DBS, another well-established intervention for Parkinsonian tremor. Given that both GPi and STN play key roles in tremor generation, their omission raises questions about whether the proposed "Convergent Mapping of a Tremor Treatment Network" fully accounts for treatment effectiveness. Additionally, GPi and STN DBS provide broader benefits beyond tremor control, improving other Parkinsonian symptoms—an advantage VIM-DBS does not achieve to the same extent.

In summary, while your study convincingly supports a common tremor network, it does not fully establish it as a universal therapeutic target, as effective interventions may act on external circuits rather than the tremor network itself. This principle may extend to other tremor types, such as dystonic and Holmes tremor, further emphasizing the need for tailored therapeutic approaches.

Nevertheless, this manuscript undoubtedly deserves acceptance for publication. Future research should continue exploring novel and more effective DBS targets to enhance tremor control, minimize adverse effects, and improve overall patient outcomes. Clarifying these distinctions would further strengthen the manuscript's impact and its contribution to the field.

Reviewer #3

(Remarks to the Author)

NO further comments

Convergent causal mapping of a tremor treatment network
(Goede et al.) – Response to Referees

Legend:

Reviewer Comment

Author Response

Changes in Manuscript

We would like to thank the reviewers for their thoughtful comments.

Reviewer #1

Remarks to the Author:

1. There is an extra comma between “neuromodulation” and “in the future” on Line 61.

Thank you very much for bringing this to our attention. We removed the comma.

2. There is an extra comma between “mapping” and “studies” on Line 94.

Thank you for pointing this out. We have revised the sentence to enhance clarity. The updated version reads:

“Studies using a related approach, known as DBS network mapping, have identified relevant networks associated with overall clinical improvement in both PD and ET patients ^{21,22}.” – introduction, p. 3

3. If DBS network maps from previous studies across various targets and disorders already indicate tremor benefits, can the authors elaborate on the additional clinical advantages of producing a “multimodal tremor network” that combines data from functional MRI, atrophy, and DBS studies?

We appreciate this important question. We see the following points:

- Neuroscientifically, it is key to demonstrate that tremor in different disorders (PD and ET) as well as the ones modulated with different DBS targets (VIM, STN, GPi) map onto a shared network. Such basic science findings are key to advance our neuroscientific knowledge, constrain the hypotheses we may generate and, ultimately, lead to clinical insights, too:
- Clinically, this knowledge can
 - lead to mechanistic understanding of why, for instance, VIM-DBS is effective for both tremor in ET and tremor in PD.
 - inform noninvasive forms of neuromodulation by reinforcing the cortical sites associated with tremor improvements (for practical examples, see Fox et al. 2014 PNAS, the entire field of TMS for depression and our recent noninvasive clinical trial for PD in Goede et al. 2024 Movement Disorders).
 - Adding robustness: By integrating data from functional MRI, atrophy maps, and DBS studies across various conditions, we aim at constructing a network that is robust to patient and disease heterogeneity. If a network is calculated only on a single modality (e.g. STN-

DBS for PD), it will almost certainly not generalize well to other forms of treatment (VIM-DBS for ET), or other modalities (fMRI, atrophy network mapping, etc.). Hence, the included heterogeneity is not a limitation but was a deliberate study design choice. By studying a diverse cohort and multiple modalities, we aim to ensure our findings are broadly applicable.

We see additional impact in our approaches to map the brain using causal sources of information (lesions, brain stimulation targets, etc): For example, previous studies have demonstrated that lesions from different etiologies causing the same symptom map to the same brain circuit, as do lesion and atrophy locations, and even invasive and noninvasive stimulation sites targeting different brain regions but modulating the same symptom (e.g. Siddiqi et al., Nat Hum Behav 2021; Schaper et al. 2023 JAMA Neurol). Additionally, DBS targeting distinct brain areas has been shown to converge on the same circuit for tremor relief. These findings collectively underscore the feasibility and potential of a unified, multimodal tremor network.

Looking ahead, this approach has the potential to inform a range of tremor-related conditions beyond essential tremor or Parkinson's disease, provided we can access larger, more diverse datasets.

We added the following paragraph to further clarify impact to the discussion session:

“In general, functional networks derived from single disorders or modalities are limited in their applicability and often fail to translate across diseases or neuromodulation approaches. For instance, while DBS for essential tremor may benefit from targeting a specific circuit, it is unlikely to inform transcranial electric stimulation strategies for Parkinson's disease. A multimodal approach as taken here, which integrated data from functional MRI, atrophy maps, and DBS studies may provide a more robust network capable of addressing patient and disease heterogeneity and generalizing to unseen modalities or disorders. By embracing this heterogeneity, we aimed to ensure our findings will be broadly applicable to various tremor-related disorders and stimulation modalities. Importantly, the principles of mapping circuits using various causal sources of information (referred to as convergent causal mapping²⁴) extend beyond tremor disorders. For example, in neuropsychiatric disorders, a treatment network overlap has been identified between obsessive-compulsive disorder (OCD) and obsessive-compulsive behavior in Tourette's Syndrome^{45,46}. Moreover, networks related with tic-reduction in secondary tic disorders and Tourette's Syndrome aligned, as well^{47,48}. Networks associated with depression have been shown to be shared across Major Depressive Disorder, Parkinson's Disease, epilepsy, multiple sclerosis and stroke^{24,49}. These findings further validate the potential of symptom-specific brain networks to inform therapeutic strategies *across diverse conditions*, underscoring the utility of a multimodal approach to neuromodulation⁵⁰.” – discussion, p. 13

4. In the results section, the authors report that the degree of spatial similarity between a stroke-lesion-based tremor-relief map and a connectivity map derived from VTA correlates with relative improvement.

5. For the above point (Fig. 1), could the authors provide an example of:

- (a) A VTA-derived connectivity map (seed map) that exhibits high spatial similarity with the stroke-based lesion map.
- (b) A map with low spatial similarity as a reference.

Thank you for this comment, this will add helpful information to this figure. We added one example with high connectivity to the lesion network map and one example with low connectivity to the lesion network map in panel C of Fig. 1 and mark the two examples in the associated scatter plot.

Fig. 1. Lesion-derived tremor-relief map connectivity correlates with clinical improvement in deep brain stimulation (DBS) cohorts. **A:** The panel illustrates 11 published cases of patients who experienced tremor alleviation following beneficial lesions due to stroke¹⁸. Using lesion network mapping, a tremor-relief network was identified (bottom). Colors represent the probability (p) of voxels associated with tremor relief. **B:** Electrode localizations from subjects in the current study, with blue representing VIM patients and orange representing STN patients. The bottom plot shows a positive correlation between relative clinical improvement (measured as ratios of the maximum achievable score for UPDRS-III tremor scores or FTM) following DBS and each DBS electrode's connectivity to the lesion network map. A cohort regressor was applied to control for potential confounding effects related to differences between patient groups. The red circle highlights an example of a volume of activated tissue (VTA) with low connectivity, shown alongside an example with high connectivity (green circle) to the lesion network map in panel **C**.

6. It appears that the stroke-lesion-based tremor-relief map represents significance (p -value) of tremor relief, while the VTA-derived seed map depicts average connectivity coefficients (r -value). Given that these are distinct statistical metrics, how was spatial similarity computed between the two maps?

Thank you for this question. The connectivity between the lesion network map and each VTA-derived seed map was calculated using voxel-wise multiplication. This approach produces a metric that reflects the degree of connectivity between two seeds. The lesion network map was generated in the original work (Joutsa et al., Ann. Neurol., 2018) by identifying and overlapping all voxels functionally connected to lesions associated with tremor alleviation. The resulting map provides a value for each voxel, indicating the probability that it is connected to these lesions. For instance, a probability value of 0.9 signifies that 90% of the 11 analyzed lesions were connected to the voxel. The map was derived 'as is' from the original report and we use the same metric as used in the original paper. To clarify this, we have added the following explanation of the lesion network map creation to the methods section:

"This map includes values for each voxel, representing the overlap of voxels functionally connected to all 11 lesion locations. A higher number of lesions connected to a specific voxel corresponds to a higher probability of tremor alleviation for the specific voxel." – methods, p. 16

7. In Fig. 1B and Fig. 2B, the observed trend appears relatively weak ($r = 0.25$ for both Fig.), though supported by a larger sample size through the combination of data from VIM and STN DBS targets. Do similar results hold when the analysis is performed separately for VIM and STN? Alternatively, would a multivariate regression analysis with STN/VIM grouping as a covariate provide deeper insights?

Thank you very much for suggesting these different approaches. In the current analysis, we regressed out grouping effects using a dummy variable (categorical indicator that differentiates the VIM and STN cohorts), i.e. did something very similar to the second suggestion by the reviewer, already. When instead adding in the dummy grouping variable into a linear model (multiple regression), the model remained significant: $R^2 = 0.135$, $F(3, 133) = 6.91$, $p < 0.001$.

As suggested, we also performed additional analyses for the VIM and STN cohorts:

For the lesion-derived map, shown in Fig. 1, the additional analysis indicates that the correlation between connectivity to the lesion map and clinical improvement is significant in the VIM cohort ($r = 0.29$, $p = 0.015$) but was not significant in the STN cohort ($r = 0.15$, $p = 0.211$).

For the EMG-fMRI derived map, shown in Fig. 2, the correlation between connectivity to the EMG-fMRI derived map and clinical improvement is positive but misses statistical significance in the VIM cohort ($r = 0.21$, $p = 0.074$), while being positive and significant in the STN cohort ($r = 0.24$, $p = 0.046$).

These results were added to the results section as follows:

“Similarity coefficients between the published a-priori tremor-relief map that was based on lesion network mapping of eleven cases that had tremor alleviation following a stroke¹⁸ and the connectivity maps derived by volumes of activated tissue (VTA) around the DBS electrodes significantly correlated with empirical clinical improvements of DBS patients ($R = 0.25$; $p = 0.003$; Fig. 1). This was done after regressing out cohort grouping (STN-DBS vs. VIM-DBS) from connectivity estimates. When instead accounting for cohort grouping in a joint linear model, this remained significant: $R^2 = 0.13$, $F = 10.27$, $p = 10^{-4}$. This finding validated the a-priori lesion map using DBS and suggests that a therapeutic target from tremor is a brain circuit that shares topography across multiple lesion locations, multiple DBS sites, and multiple underlying diagnoses to alleviate tremor. When repeating the analysis for each DBS cohort alone, clinical improvements correlated positively with connectivity to the lesion network map in both cohorts, but this association was only significant for VIM-DBS for ET ($R = 0.29$, $p = 0.015$) and not for STN-DBS for PD ($R = 0.15$, $p = 0.211$).” – results, p. 5

“A third analysis was conducted with a published network map that had been established using a concurrent EMG-fMRI experiment on 22 patients with PD, which led to the influential ‘dimmer-switch’ model of tremor²⁷. This analysis also revealed a significant association between connectivity to the fMRI tremor activity map and clinical improvements ($R = 0.22$; $p = 0.013$; Fig. 2). As above, this analysis was carried out after regressing out cohort grouping (STN-DBS vs. VIM-DBS) from connectivity estimates. When instead accounting for cohort grouping in a joint linear model, this remained significant: $R^2 = 0.13$, $F = 9.69$, $p = 10^{-3}$. When repeating the analysis for each DBS cohort alone, clinical improvements correlated positively with connectivity to the lesion network map in both cohorts, but this association was only significant for STN-DBS for PD ($R = 0.24$, $p = 0.046$) and not for VIM-DBS for ET ($R = 0.21$, $p = 0.074$).” – results, p. 6

We added a paragraph in the discussion section of our manuscript:

“Several limitations should be considered when interpreting our findings. First, the use of data from different patient populations for lesion network mapping and DBS analyses introduces variability. Although we focused on identifying commonalities across modalities, differences in disease progression, specific symptom components, treatment history, and patient characteristics may impact the generalizability of our results. Furthermore, the inherent variability in clinical scales across cohorts may also limit the interpretation of findings. However, this heterogeneity was deliberately chosen to increase robustness of findings. As outlined in previous publications, any network conclusion that is primarily based on a single dataset of a single nature (e.g. STN-DBS alone) would be inherently biased towards this type of data²⁰. Previous studies in different symptoms could illustrate that the convergent analysis of heterogeneous datasets typically improve robustness and generalizability of findings^{23,24,48,50}.” – discussion, p. 14

8. As noted by the authors, habituation is reported in VIM-DBS for ET patients, which might otherwise diminish tremor benefits at longer postoperative time points. Did the authors account for the time elapsed since surgery in analyses correlating postoperative tremor benefits with connectivity maps?

Thank you for raising this important point. The mean and median follow-up time for the evaluation of the scores in the VIM-DBS cohort are both 12 months. To investigate whether longer follow-up times would influence the connectivity profile of the group map, we conducted an additional analysis. Specifically, we split the VIM-DBS cohort by the median into two subgroups: patients with a follow-up time longer than 12 months (>12 months) and those with a shorter follow-up time (<12 months). We then calculated optimal connectivity profiles for each subgroup and generated two separate maps for comparison.

The following section was added:

“To assess whether the duration of follow-up in VIM-DBS patients influenced the group map, we divided the cohort into two subgroups: patients with follow-up periods longer than 12 months (N = 10; >12 months) and those with shorter follow-up periods (N = 12; <12 months). Correlation maps of optimal clinical response still highlighted the same general profile with positive connectivity to motor strip and motor cerebellum (Fig. S2).” – supplement, p. 12

A supplementary Fig. (S2) illustrating these results has been added:

Fig. S2. Correlation maps for VIM-DBS patients based on follow-up duration. **A:** Correlation map for VIM-DBS patients with follow-up durations less than 12 months (mean follow-up: 4.42 ± 2.26 months). **B:** Correlation map for VIM-DBS patients with follow-up durations greater than 12 months (mean follow-up: 21.1 ± 13.86 months). Both maps still highlighted the same general profile with positive connectivity to motor strip and motor cerebellum.

9. In disorders like Parkinson's Disease (PD) or Essential Tremor (ET), numerous studies have demonstrated differences in fMRI/BOLD signals compared to healthy cohorts. Given that this analysis utilized group-level resting-state fMRI data from a healthy subject group, how did the authors account for disease-specific changes in BOLD activity independent of stimulation effects? Are there fMRI-based connectomes specific to ET and PD cohorts that could be used to replicate the findings?

We tried our very best to find an ET connectome but unfortunately failed. However, a PD connectome could be generated from data from the *Parkinson's Progression Markers Initiative (PPMI)*. We have still used the PD connectome to replicate main analyses, which showed high similarity to the results generated with the normative (healthy) connectome.

The following paragraphs and Fig.s were added to the manuscript:

“To evaluate the replicability of our findings, we repeated the analysis using a disease-specific connectome derived from MRI data of 90 patients from the Parkinson’s Progression Markers Initiative (PPMI) database ^{21,59}.” – methods, p. 16

“Our analyses applied normative connectome data acquired in healthy subjects. While a disease-matched connectome acquired in ET patients was not available, we replicated main results using a disease-matched connectome acquired in PD patients. This confirmed that optimal response networks remained consistent when using a PD connectome, suggesting that results were not substantially biased by the connectome itself. Furthermore, predictive performance improved for the out-of-sample cohort test, yielding an enhanced outcome estimation in the GPi-DBS PD cohort ($R = 0.62$, $p < 0.001$; Fig. S8).” – results, p. 9f.

Fig. S8. Replication of main results using a disease specific connectome derived from the Parkinson’s Progression Markers Initiative (PPMI) instead of the normative connectome derived from healthy subjects. A & B: Correlation maps representing optimal clinical improvement for STN-DBS patients and VIM-DBS patients, respectively (compare Fig. 3). **C:** Agreement map consisting of STN-DBS- and VIM-DBS-correlation maps anticipates clinical improvement in VIM- and STN-DBS-patients (compare fig. 4). **D:** Multimodally-informed convergent tremor map consisting of lesion-derived fMRI-map, EMG-fMRI-derived map, VIM-STN (agreement) map and ET-specific atrophy-map anticipates clinical outcomes in out-of-sample GPi cohort (compare fig. 5).

10. In the discussion, the authors state that “both rest and action tremors may unfold within the same anatomical network.” To better support this claim, the authors might consider using the “convergent tremor map” and quantifying connectivity to this map. This could then be correlated with not only total UPDRS-III tremor improvement but also with its action, postural, and rest tremor subscores.

Thank you for this thoughtful suggestion, which we realized in the following. In addition, we also created other maps using the different types of tremor. However, since most cohorts had been phenotyped with the original UPDRS-III tremor score (not with the novel MDS UPDRS-III), in which action and postural tremor are subsumed under the same item, we analyzed rest vs. action & postural tremor in all analyses.

The following sections were added:

“To test whether different forms of tremor would map to the same network, we repeated this analysis only using rest vs. action/postural tremor items (Fig. S6 & S7).” – results, p. 9

“Additionally, repeating analyses for different forms of tremor within the out-of-sample cohort suggested a shared network might be involved onto which these tremors unfold.” – discussion, p. 12

Fig. S6. Multimodally informed convergent tremor map estimates clinical improvement in tremor subscores. **A:** Multimodally-informed convergent tremor map, as shown in Fig. 5. **B:** Out-of-sample analysis of GPi connectivity to the convergent tremor map, with tremor improvement assessed using subscores for rest tremor and the combined action/postural tremor score. Connectivity to the convergent tremor map significantly correlated with improvement in both rest and action/postural tremor subscores.

Fig. S7. Comparison of tremor subtypes. Correlation maps representing networks for optimal clinical response demonstrate similarities across DBS target regions, disease types, and tremor subtypes (rest, action, and posture tremor). The maps highlight overlapping areas of connectivity associated with

effective tremor suppression, suggesting a shared therapeutic network underlying tremor relief irrespective of these variables.

Reviewer #2

Remarks to the Author:

The manuscript establishes a foundation by addressing the diversity of tremor-related disorders, such as Parkinson's Disease (PD) and Essential Tremor (ET), characterized by rhythmic, involuntary oscillations. While these disorders have distinct origins, the authors hypothesize a shared network underpinning tremor manifestations, a premise this study seeks to investigate. Traditional therapeutic approaches have targeted specific brain regions (e.g., the ventrointermediate nucleus (VIM) and the posterior subthalamic area (PSA)). However, emerging research proposes that tremor may arise from pathological activity within a distributed network rather than isolated locations. The authors aim to map a convergent network for tremor treatment by integrating lesion mapping, atrophy patterns, task-based fMRI data, and deep brain stimulation (DBS) network data. This ambitious approach seeks to uncover a unified treatment framework spanning various tremor types and DBS targets.

1. Combining PD and ET Populations: While the network-based approach is commendable, does it sufficiently justify merging PD and ET populations with potentially distinct etiologies and network origins? Specifically, as the results and title suggest a causative network, is it appropriate to generalize across these tremor types?

Thank you for raising these important questions. We must emphasize that cohorts were analyzed separately in multiple analyses (e.g., see fig. 3) and independently pointed to the same network, i.e. we only pooled across disorders in some analyses. This was a deliberate choice and the key aim of the entire project: Indeed, there are many pathophysiological differences between ET and PD, e.g. the additional role of the basal ganglia in PD (Helmich et al., *Mov Dis* 2018), and the more dominant role of the cerebellum in ET (Buijink et al., *Curr Opin Neurology* 2022). At the same time, in both tremor disorders the cerebello-thalamo-cortical circuit is a common pathway involved in producing the tremor (Buijink et al., *Curr Opin Neurology* 2022). Hence, we've built the hypothesis that different pathophysiological mechanisms ultimately result in a common network, leading to this project. We believe the heterogeneity of our cohorts strengthens our analysis by highlighting shared features across multiple tremor disorders and DBS targets. To reflect this, we have revised the title to remove the term "causal" to avoid potential overgeneralization and emphasize a more cautious interpretation of our findings.

The revised title now reads:

"Convergent mapping of a tremor treatment network"

2. Etiology-Specific Networks: Shouldn't the study seek to identify distinct networks for tremor based on etiology? While rhythmic oscillatory patterns may reflect similar phenomenology, PD tremor typically occurs at rest, whereas ET tremor manifests during voluntary movement. Could analyzing these disorders together obscure critical distinctions and introduce confounding variables?

Thank you for raising this important point. As mentioned above, we must emphasize that ET and PD cohorts had been analyzed separately, already in the original manuscript (e.g. see fig. 3). Both cohorts *independently* showed the same tremor network. We apologize that this had not been clear enough and added the following sections to further clarify this point. Only in some analyses did we pool across disorders, given that the key aim of the paper is to demonstrate shared outcomes.

The following sections were added:

"To explore the differences and similarities of optimal network profiles in STN- and VIM-DBS, we next tested whether one map could predict ranks in outcomes of the respective other cohort. Again, each *seedmap* was spatially correlated with the *R-map* of the respective other cohort and coefficients were subsequently correlated with clinical outcomes." – methods, p. 17

"While data-driven maps from STN and VIM DBS cohorts looked similar by visual inspection (Fig. 3), we empirically tested their similarity in two ways. First, we used one of the maps to account for variance in clinical outcomes in the respective other cohort (VIM-DBS map predicts STN-DBS patients: $R = 0.24$; $p =$

0.049; STN-DBS map predicts VIM-DBS patients: $R = 0.31$; $p = 0.009$; Fig. 3A). Second, we tested whether similarities across the two maps, as expressed by voxel-wise spatial correlations, would be higher than what could be expected by chance. To do so, we permuted outcomes in one cohort and each time compared the similarity of the resulting R -map with the respective other (unpermuted) R -map to generate null-models of similarities (10,000 iterations). In both directions, similarities of unpermuted R -maps were significantly larger than expected by chance (permuting STN-DBS: $p = 0.02$; permuting VIM-DBS: $p = 0.03$; permuting both cohorts at the same time: $p = 0.02$; Fig. 3B).” – results, p. 7.

Fig. 3. Cross-validation and permutation analysis between DBS cohorts. A: Correlation maps for both the STN-DBS cohort (top left) and the VIM-DBS cohort (bottom right). These maps contain group-level correlation coefficients consisting of each patient’s hemisphere connectivity profile with their clinical tremor improvement. The connectivity of VIM-DBS patients to the STN connectivity profile correlated with clinical improvement (top), and vice versa for STN patients with VIM profiles (bottom). **B:** Results of a permutation analysis for VIM connectivity maps predicting STN-DBS outcomes (left), STN connectivity maps predicting VIM-DBS outcomes (right), and both combined (middle).

“Critically, the same tremor network resulted from independently analyzing PD and ET cohorts that underwent DBS, i.e., the shared topology of the network did not result from pooled data analysis.” – discussion, p. 11

Additionally, we performed a focused analysis to explore features unique to each disorder, specifically rest tremor (associated with Parkinson’s disease) and action/postural tremor (typical of essential tremor). The results of this analysis are detailed in the supplementary Fig. S6. This approach allowed us to balance the identification of commonalities across etiologies while preserving insights into their distinct characteristics.

The following sections were added:

“To test whether different forms of tremor would map to the same network, we repeated this analysis only using rest vs. action/postural tremor items (Fig. S6 & S7).” – results, p. 9

“Additionally, repeating analyses for different forms of tremor within the out-of-sample cohort suggested a shared network might be involved onto which these tremors unfold.” – discussion, p. 12

Fig. S6. Multimodally informed convergent tremor map estimates clinical improvement in tremor subscores. **A:** Multimodally-informed convergent tremor map, as shown in Fig. 5. **B:** Out-of-sample analysis of GPi connectivity to the convergent tremor map, with tremor improvement assessed using subscores for rest tremor and the combined action/postural tremor score. Connectivity to the convergent tremor map significantly correlated with improvement in both rest and action/postural tremor subscores.

Fig. S7. Comparison of tremor subtypes. Correlation maps representing networks for optimal clinical response demonstrate similarities across DBS target regions, disease types, and tremor subtypes (rest, action, and posture tremor). The maps highlight overlapping areas of connectivity associated with effective tremor suppression, suggesting a shared therapeutic network underlying tremor relief irrespective of these variables.

3. Hypotheses on Network Convergence: The introduction alludes to shared tremor-relief networks across conditions. How do the authors specifically hypothesize these networks will converge or diverge between disorders?

Thank you for your question regarding our hypotheses on network convergence. In our study, we hypothesize that tremor-relief networks across different disorders (e.g., PD and ET) converge within a shared symptom-specific network. Specifically, we propose that this network involves connections between the thalamus, cerebellum, and primary motor cortex, as supported by recent studies using lesion network mapping, DBS network mapping, atrophy mapping, and EMG-fMRI research.

We further hypothesize that the extent of convergence or divergence within these networks depends on the specific characteristics of each disorder. Core components of the tremor-relief network, such as the motor cerebellum and primary motor cortex, may serve as universal hubs across disorders, reflecting their fundamental roles in tremor modulation (convergence). Conversely, disorder-specific variations may arise due to distinct etiologies and pathophysiological mechanisms. For example, cerebellar degeneration in ET versus basal ganglia dysfunction in PD, potentially influencing the strength of specific pathways within the shared network (divergence). These hypotheses are informed by recent publications that highlight both shared and disorder-specific features in tremor-relief networks. Finally, a shared anatomical topography does not imply that the dysfunctional oscillations that unfold across the networks are the same. From clinical observations, for instance, we know that rest tremor occurs at rest while action tremor occurs while carrying out actions. So, across the temporal domain, signatures of these two distinct symptoms will certainly be different – but may still unfold across a shared anatomical network.

To better address this point in our manuscript, we have refined the final paragraph of the introduction as follows:

"In tremor, DBS network mapping identified a set of brain regions^{22,23} very similar to those found in lesion network mapping studies reported earlier¹⁸, primarily involving the primary motor cortex and motor parts of the cerebellum. A recent study that subjected atrophy patterns to network mapping analyses found yet again a key hub in overlapping (motor) regions of the cerebellum²⁶. Additionally, studies using concurrent electromyography (EMG) and functional MRI (fMRI) have also explored tremor-related activity in the brain and identified yet again a shared set of connections between thalamus, cerebellum and primary motor cortex with tremor^{27,28}. These findings support the same notion that the optimal target for tremor relief is not a single brain region but rather a network that may converge on shared hubs while allowing for disorder specific variations. Therefore, comparing and integrating results from lesion network mapping, functional MRI, atrophy, and DBS network mapping could yield a unified map of a tremor treatment network. However, no study has yet analyzed converging evidence from these maps and set the resulting network pattern into relationship with tremor outcomes across disorders (ET and PD) or across different DBS targets (VIM, STN, and GPi). Here, we aim to address this gap by first investigating whether networks identified in lesion network mapping, EMG-fMRI and atrophy network mapping studies can explain clinical improvements in large cohorts of patients who underwent DBS surgery for PD and ET targeting the STN and VIM. Second, we develop a data-driven tremor network using these DBS cohorts to determine whether they independently identify similar or distinct networks across tremor disorders and DBS targets. Third, we integrate these findings with a priori results from lesion network mapping, fMRI and atrophy network mapping to generate a multimodal tremor network. Finally, we test whether this multimodally informed network may account for improvements in tremor in a third large cohort of PD patients that underwent DBS to the GPi, as a third DBS target." – introduction, p. 4

"A shared anatomical topography does not imply that the dysfunctional oscillations that unfold across the networks are the same. From clinical observations, for instance, we know that rest tremor occurs at rest while action tremor occurs while carrying out actions. So, across the temporal domain, signatures of these two distinct symptoms will certainly be different – but may still unfold across a shared anatomical network." – discussion, p. 12

4. Details of Task-Based fMRI: The manuscript references task-fMRI maps but does not specify the motor tasks involved. This is critical because nearly all motor tasks engage the cerebello-thalamo-cortical circuit, widely recognized as the final common pathway for movement. Are the tasks designed to elicit tremor-related BOLD activity? Details on rest vs. action tremor paradigms, specific motor events, and the participant populations (PD and ET patients or healthy controls) should be explicitly clarified.

Thank you for pointing this out. The term "task-fMRI" written in our manuscript was indeed confusing. What we have done is to use concurrent EMG and fMRI scanning to measure fluctuations in PD resting tremor amplitude during scanning and relate these fluctuations to BOLD fluctuations. Given the EMG data was used to define temporal points in time, typical task-fMRI analysis pipelines were used, but patients did not actually carry out a task. The cohort consisted of 22 tremor-dominant PD patients who all had resting tremor during scanning.

More specifically, as outlined in the paper from which the “fMRI tremor map” (Fig. 2) was derived (Dirkx et al., J Neuroscience 2016), we have now added the description of the methods to the supplement, and we have changed “Task-Based fMRI” to “EMG-fMRI” throughout the manuscript:

“The cohort consisted of 22 tremor-dominant PD patients who all had resting tremor during scanning¹⁷. The following steps were performed that led to the “fMRI tremor map” (Fig. 2):

Muscle activity of the most affected forearm (wrist flexors and extensors) was measured using MR-compatible EMG (Brain Products; sampling frequency 5000 Hz) during fMRI scanning in all 22 patients. BrainVision Analyzer 2.0 (Brain Products) was used for preprocessing the EMG data. Then we: (1) used MR artifact correction¹⁸, (2) down-sampled to 1000 Hz, (3) filtered with a 20-200 Hz band-pass filter to remove movement artifacts, and (4) rectified the signal to enhance the information on EMG burst-frequency (tremor) of the signal, thereby also recovering the low-frequency EMG content¹⁹. Next, EMG data were analyzed using FieldTrip²⁰. Specifically, we calculated the time-frequency representations (TFRs) between 1 and 20 Hz in steps of 0.1 s using a 2 s Hanning taper, which resulted in a 0.5 Hz resolution. By averaging over all time points, we obtained an average power spectrum across segments. For each patient, we picked the TFR of the corresponding tremor frequency (i.e., the peak in the power spectrum between 4 and 6 Hz; mean: 4.6 ± 0.1 Hz), resulting in patient-specific regressors describing fluctuations in tremor amplitude (EMG-AMP). To remove outliers, we calculated the logarithmic values of the EMG-AMP and z-normalized the data within subjects. Finally, the EMG-AMP regressor was convolved with the hemodynamic response function, and the resulting regressor was added to our first-level model in SPM, next to several regressors of no interest: two regressors describing the signal intensity averaged on each scan over the segmented gray matter (i.e., global signal, to correct for head motion²¹) and over a blank portion of the MR images (out-of-brain signal) and 36 regressors describing head motion. Regressors describing head motion were based on linear, quadratic, and cubic effects of the six movement parameters belonging to each volume, as well as the first and second derivative of each of those regressors (to control for spin-history effects²²).

The “fMRI tremor map” (Fig. 2) are voxels where BOLD activity is significantly associated with fluctuations in tremor power during scanning, averaged across 22 tremor-dominant PD patients. This approach has been done and replicated in several studies, all showing the same cerebello-thalamo-cortical circuit^{23–26}.”
– supplementary material, method description S1, p. 13

5. fMRI method: Would fMRI be considered a good method for specifying network activity related to an oscillatory phenomenon once BOLD effect is an indirect method to show brain regional activation and also features low temporal resolution?

Thank you for this question. Tremor is indeed an oscillatory phenomenon with a frequency between 4-6 Hz in PD, and in ET usually between 4-8 Hz. Functional MRI does not have the temporal resolution to detect these cycle-by-cycle oscillations. However, PD tremor also exhibits much slower fluctuations in power, which occur over multiple seconds (i.e. the “envelope” of the oscillatory signal). This is clinically seen as a very characteristic “waxing and waning” of resting tremor amplitude (Dirkx et al., Mov Dis 2023, Zach et al. J Parkinson’s disease 2015). Functional MRI is very sensitive to these slower fluctuations in tremor power. Using the EMG-fMRI approach described above, here we leveraged this phenomenon to map tremor-related brain regions. EEG and MEG approaches, which focus on the cycle-by-cycle 4-8 Hz rhythmic activity, show brain circuits that are very similar to the EMG-fMRI tremor maps (Helmich et al., Brain 2012, Timmermann et al., Brain 2003), but these methods are hampered by a lower spatial resolution in subcortical brain areas that are relevant to tremor (such as the VIM).

To clarify this further, we adjusted the following paragraph in the discussion section of our manuscript:

“Third, while fMRI data offers valuable insights, its relatively slow temporal resolution does not capture the rapid dynamics of neural activity associated with tremor. Nonetheless, our findings agree with electrophysiological data, which offer complementary evidence of the neural mechanisms at play^{34,35}. Indeed, they align with EEG and MEG approaches that focus on the cycle-by-cycle 4–8 Hz rhythmic activity, although these methods are hampered by a lower spatial resolution in subcortical brain areas relevant to tremor, such as the VIM^{37,51}. Integrating connectomic analysis with these electrophysiological findings and

prior research may provide a more comprehensive understanding of tremor networks, in the future.” – discussion, p.14

The study includes PD and ET patients undergoing DBS targeting the STN, VIM, and GPi across multiple international centers. Lesion maps, atrophy patterns, and task-fMRI maps were used as reference networks to assess connectivity across these DBS targets. DBS electrode locations and volumes of tissue activated were modeled using the Lead-DBS software. Data-driven maps derived from each cohort were cross-validated and integrated to create a unified, multimodal "agreement map." Statistical analyses primarily relied on Spearman's rank correlations and permutation tests.

1. Sample Size Adequacy: Are the sample sizes for each tremor population sufficient to support the observed correlations and ensure generalizability?

Thank you, this is an important point. Alongside the manuscript we submitted a "Reporting summary form" that describes our power analysis, and will be published:

"Given the exploratory nature of our study, conducting an a priori power analysis was challenging. We based our expected effect size on Al-Fatly et al. (Brain, 2019), with an $r = 0.36$ for reported correlations. Considering the limitations in available sample sizes across DBS targets, we performed a "compromise" power analysis using G*Power Version 3.1.9.6 (Faul et al., 2007; 2009, Behav. Res. Methods) to estimate the power of our analysis. We assumed equal Type I (α) and Type II (β) error probabilities, with a ratio of 1. The power estimates were as follows: VIM cohort ($n = 72$ hemispheres) had a power of 0.93 (α/β error probability = 0.07), STN cohort ($n = 65$ hemispheres) had a power of 0.91 (α/β error probability = 0.09), and GPi cohort ($n = 31$ hemispheres) had a power of 0.80 (α/β error probability = 0.20). These power values suggest that the VIM and STN cohorts had strong statistical power to detect the expected correlations, while the GPi cohort demonstrated adequate power for reliable detection." – reporting summary form, p. 2

2. Merging Data Across Disorders: The integration of PD and ET datasets for network analysis assumes shared pathophysiology. However, combining these populations risks oversimplifying distinct disease mechanisms. Would separate analyses yield more nuanced insights?

We believe this point is redundant with the first two points raised by the reviewer and we have addressed them above. Again, we must emphasize that i) cohorts were only pooled for some analyses and ii) this was the general aim and hypothesis of the paper. We hope that this point could be clarified by our extensive response to the second query by the same reviewer.

3. Normative vs. Individual Connectivity Variability: Given that Lead-DBS and its associated connectome rely on normative data, could individual patient variability in connectivity affect reproducibility?

While we lack individual fMRI data from these DBS cases, we now substituted the normative connectome using a disease-specific PD connectome derived from the *Parkinson's Progression Markers Initiative (PPMI)* and repeated key analyses, showing that the choice of connectome did not have an impact on main results.

The following paragraphs were added to the manuscript:

"To evaluate the replicability of our findings, we repeated the analysis using a disease-specific connectome derived from MRI data of 90 patients from the Parkinson's Progression Markers Initiative (PPMI) database^{21,59}." – methods, p. 16

"Our analyses applied normative connectome data acquired in healthy subjects. While a disease-matched connectome acquired in ET patients was not available, we replicated main results using a disease-matched connectome acquired in PD patients. This confirmed that optimal response networks remained consistent when using a PD connectome, suggesting that results were not substantially biased by the connectome itself. Furthermore, predictive performance improved for the out-of-sample cohort test, yielding an enhanced outcome estimation in the GPi-DBS PD cohort ($R = 0.62$, $p < 0.001$; Fig. S8)." – results, p. 9f.

Fig. S8. Replication of main results using a disease specific connectome derived from the Parkinson's Progression Markers Initiative (PPMI) instead of the normative connectome derived from healthy subjects. A & B: Correlation maps representing optimal clinical improvement for STN-DBS patients and VIM-DBS patients, respectively (compare fig. 3). **C:** Agreement map consisting of STN-DBS- and VIM-DBS-correlation maps anticipates clinical improvement in VIM- and STN-DBS-patients (compare fig. 4). **D:** Multimodally-informed convergent tremor map consisting of lesion-derived fMRI-map, EMG-fMRI-derived map, VIM-STN (agreement) map and ET-specific atrophy-map anticipates clinical outcomes in out-of-sample GPi cohort (compare fig. 5).

The results indicate varying levels of tremor improvement across DBS targets, with STN-DBS (PD) showing 87.4% improvement, VIM-DBS (ET) 67.0%, and GPi-DBS (PD) 90.1%. Significant associations were observed between tremor improvement and connectivity to lesion-based tremor relief maps and fMRI tremor maps, but not with atrophy maps. Cross-validation between STN and VIM cohorts revealed predictive capabilities, suggesting an overlapping network. However, connectivity-to-outcome correlation coefficients were relatively low ($R=0.11$ to $R=0.45$).

1. Predictive Power of Connectivity Studies: Low correlation coefficients could indicate weak predictive power for connectivity studies in forecasting tremor improvement. Would refining these models improve their reliability?

Thank you for raising this important point. While the observed effect sizes may appear low, it is essential to emphasize that clinical outcomes are influenced by numerous factors beyond electrode location alone (e.g. patient age, comorbidities, baseline tremor severity, etc). However, a key point is that electrode localization is one of the few parameters we can actually influence during clinical care. Hence, it is a key parameter to optimize. We include novel Fig. S1 (pasted below) to visually describe and discuss this issue.

As noted in our discussion, we accounted for heterogeneity by integrating multimodal data across DBS targets and performing separate analyses to clarify both shared and target-specific network components. To our knowledge, this represents the first effort to combine lesion mapping, atrophy patterns, task-fMRI, and DBS data across three different DBS targets, resulting in a robust and generalizable model.

We added a paragraph in the limitations section of our manuscript:

“While our linear regression model accounted for cohort differences and their interaction with connectivity, it is important to note that the model explained only a modest portion of the variance in clinical improvement (adjusted $R^2 = 0.13$ for the lesion derived map and $R^2 = 0.13$ for the EMG-fMRI derived map). This suggests that other unmeasured factors, such as individual patient characteristics, differences in disease pathology, or variability in clinical scales, may play a significant role. We approximate the impact of these factors according to literature reports in Fig. S1. Additionally, the variability in clinical scales (e.g., MDS-UPDRS-III for Parkinson's disease vs. FTM-Tremor scale for essential tremor) may influence the sensitivity of our analysis and should be considered when interpreting these results. Critically, however, none of these variables may be influenced due to medical practice, with the sole exception of the electrode placement and stimulation settings, which renders the defined tremor response target (which is based on these two factors alone) the critical anchor point with an opportunity to potentially improve patient care ²⁵.” – discussion, p. 13

Fig. S1. Modeling considerations. The pie charts depict factors influencing patient outcomes following DBS, as determined through literature review. For Parkinson's disease (PD), more studies have investigated determinants of DBS outcomes compared to other conditions. Key considerations include the variance that DBS modeling can realistically explain. Clinical improvements depend on multiple factors beyond electrode placement and stimulation volumes, such as disease subtype, age, sex, levodopa

response, disease duration, and comorbidities, which collectively account for approximately 50% of the variance in DBS response. Measurement noise, including inter- and intra-rater reliability of UPDRS-III scores, contributes another ~20%. Imaging resolution, electrode placement inaccuracies, and modeling limitations add ~10% variance in PD and ~30% in essential tremor (ET). Habituation effects are present particularly in VIM-DBS for ET. Further factors remain uncertain but may further influence outcomes. Despite these challenges, such models remain clinically valuable. Electrode placement and stimulation settings are modifiable, unlike immutable patient factors such as age or disease type. Consequently, identifying an optimal target that explains even ~10% of variance represents a meaningful advancement in the field¹⁻¹⁰.

*It is uncertain how much variance in explained clinical outcomes imaging reconstruction errors will have since no study deliberately tested for this. However, multiple reports speak of reconstruction errors of at least 1 mm displacement⁴⁻⁶. The only study that we are aware of which tested for artificially introduced spatial uncertainty imposed a jitter of N=258 electrode placements according to a 3D Gaussian distribution of 2mm full width half maximum. In this report, the standard error interval in outcome predictions for iteratively jittered group analyses ranged from ~60 % and ~70 % explained variance, i.e. a standard error of around 10%.

Additionally, a paragraph describing a performed generalized linear model (GLM) that included patient baseline scores, sex, and age has been added:

“For the ET cohort, a generalized linear model (GLM) incorporating connectivity to the optimal model correlation map, baseline scores, patient age, and sex explained a significant proportion of the variance in clinical outcomes ($R^2 = 0.35$, $F = 35.8$, $p < 0.001$). Significant predictors included connectivity values ($t = 3.46$, $p < 0.001$) and baseline scores ($t = 10.05$, $p < 0.001$), whereas age ($p = 0.18$) and sex ($p = 0.64$) were not significant. Similarly, for the PD cohort, a GLM incorporating connectivity to the optimal model correlation map, baseline scores, patient age, and sex explained a substantial amount of variance in clinical outcomes ($R^2 = 0.42$, $F = 43.9$, $p < 0.001$). Significant predictors in this cohort were connectivity values ($t = 2.94$, $p = 0.005$) and baseline scores ($t = 11.13$, $p < 0.001$), while age ($p = 0.12$) and sex ($p = 0.08$) did not reach statistical significance. Notably, none of these variables can be influenced through medical practice, except for electrode placement and stimulation settings. This renders the model estimates, derived from these modifiable factors, a critical anchor point with the potential to improve patient care (see Fig. S1).” – supplement, p. 12

2. Separate Analyses for DBS Targets: Would distinct analyses of STN, VIM, and GPi targets, as well as separate evaluations of tremor types, provide deeper insights into DBS effectiveness?

Thank you for your question, but we feel that this again taps into the same point we addressed in our response to query #2. We apologize in case we missed something here. In brief, again, all DBS cohorts were analyzed separately (e.g. figs 3 and 5).

3. Atrophy Maps and Predictive Limitations: The absence of significant associations with atrophy network maps highlights potential limitations in using this modality for tremor prediction. Could excluding these maps enhance the validity of the agreement map?

We repeated the analysis excluding the atrophy map (novel Fig. S5 pasted below. This did not change much. The following paragraph was added to the results section:

“Connectivity from DBS sites in the GPi cohort significantly correlated with connectivity to the convergent tremor map ($R = 0.45$; $p = 0.009$; Fig. 5) and remained unchanged and significant when repeating the analysis without including the atrophy map (Fig. S5).” – results, p. 9

Fig. S5. Multimodally-informed convergent tremor map without including the atrophy map anticipates clinical outcomes in out-of-sample GPI cohort. **A:** The lesion-derived map, EMG-fMRI-derived map and the VIM-STN (agreement) map were superimposed to create the displayed *multipodally-informed convergent tremor map*, with z-scores visualized. **B:** In an out-of-sample cohort of 31 analyzed hemispheres from PD patients with GPI-DBS, lead localizations were conducted using Lead-DBS (localizations shown in the top right image)¹⁵. Correlation between connectivity of DBS sites to the convergent tremor map and MDS-UPDRS-III tremor improvements.

4. Lesion Sites vs. Atrophy Maps: The findings suggest lesion sites are more predictive than atrophy maps. How does this reconcile with the hypothesis of a network origin? Furthermore, GPI targeting for PD tremor, located outside the cerebello-thalamo-cortical circuitry, produced better outcomes than VIM targeting within the hypothesized network. Do these findings fully support the proposed conclusions?

Thank you for your thoughtful question. It is important to clarify that atrophy is not causal to symptoms but rather a downstream effect, likely resulting from prolonged network dysfunction. This distinction underscores why lesion sites, which directly disrupt network connectivity, may appear more predictive in our analyses. However, our study is not powered to definitively support the assertion that “lesion sites are more predictive than atrophy maps.” To address this concern and further validate our findings, we performed an additional analysis excluding the atrophy maps (see above). The results of this analysis, presented in supplementary Fig. S2, showed that the multimodally informed convergent tremor map remained robust, with no significant changes in correlation strength or predictive validity. The Fig. is displayed above.

The authors conclude that integrating lesion mapping, atrophy patterns, task-fMRI, and DBS data supports the existence of a symptom-specific tremor network, potentially enabling treatment independent of tremor etiology. This network reportedly involves the motor cortex, cerebellum, and thalamus, suggesting a pathway for symptom-specific DBS interventions. However, they acknowledge limitations stemming from the study’s correlational design, heterogeneous populations, and multicenter data, which could introduce variability yet potentially enhance robustness.

1. Balancing Variability and Robustness: The study emphasizes the robustness achieved through diverse populations. Could stricter inclusion criteria or subgroup analyses mitigate variability and strengthen the network's validity?

Thank you for this insightful question. We agree that balancing variability and robustness is critical. To address this, as mentioned in multiple replies above, we performed separate analyses for individual DBS targets (Fig. 3 for VIM and STN separately, and Fig. 5 for GPi) to ensure our findings reflect both shared and target-specific network components. Additionally, we applied strict inclusion criteria, such as requiring a minimum tremor score of 3, to mitigate variability across datasets. To our knowledge, this study represents the first integration of such diverse datasets, including lesion mapping, atrophy patterns, task-fMRI, and DBS data (across three different DBS targets). This approach uniquely enhances the robustness and generalizability of our proposed symptom-specific tremor network.

We added a paragraph within the discussion of our manuscript:

“Our study incorporated diverse datasets across multiple DBS targets and integrated multimodal approaches, making it, to our knowledge, the first to achieve this level of comprehensive synthesis in tremor. While we deliberately embraced heterogeneity to enhance the robustness of our findings, we also applied inclusion criteria, such as requiring a tremor score above two, to reduce noise in the data. Additionally, all cohorts were also analyzed separately, and separate analyses were conducted to ensure that target-specific and shared networks were accurately identified.” – discussion, p. 14

2. Clinical Implications of Low Effect Sizes: Given the low effect sizes observed, what are the practical implications of targeting specific network hubs? Would a more focused analysis better clarify optimal DBS targets?

Thank you for this important question. While the observed effect sizes may appear low, it is essential to emphasize that clinical outcomes are influenced by multiple factors beyond electrode location, but electrode location is the critical one factor we can influence, clinically. The reviewer has already raised this exact same issue above, so for reasons of brevity, we choose to not give the same reply again here.

3. Role of Atrophy Maps: With atrophy maps showing no significant correlations, how does this influence the predictive value of the multimodal approach? Should their exclusion or refinement be considered for future research?

The reviewer has already raised this exact same issue above, so for reasons of brevity, we choose to not give the same reply again here (in brief, repeating the analysis without the atrophy map yielded the same results).

Final Concerns:

Premise and Hypothesis Misalignment

The central hypothesis of the manuscript suggests a “common and causative tremor network” rooted in the cerebello-thalamo-cortical circuit. However, this circuit is already widely acknowledged as the common final pathway for virtually all forms of movement, including tremor. The study does not present novel evidence to distinguish this network as uniquely causative for tremor over other motor phenomena.

Furthermore, while the cerebello-thalamo-cortical circuit is a logical focus for therapeutic targets such as Vim in DBS for tremor, the clinical results paradoxically demonstrate superior outcomes for STN and GPi DBS. These targets lie outside the cerebello-thalamo-cortical network and instead address pathological changes in the basal ganglia circuitry, particularly those associated with dopamine deficiency in Parkinson's disease (PD). This discrepancy undermines the conclusion that the cerebello-thalamo-cortical network is the optimal therapeutic focus for tremor relief.

Implications of the Conclusions

The conclusions, if left unrefined, may lead readers to infer that DBS targets within the cerebello-thalamo-cortical circuit (e.g., Vim) are inherently superior or more mechanistically appropriate for tremor treatment. However, the observed efficacy of STN and GPi DBS in PD tremor challenges this narrative. Future

research would benefit from focusing on DBS targets identified through mechanistic insights—such as addressing basal ganglia dysfunction in PD—rather than defaulting to the cerebello-thalamo-cortical network, which serves as a generic movement-related pathway.

Additional Limitations

Justification for Combining PD and ET Populations

The manuscript combines data from PD and essential tremor (ET) populations, which are known to have distinct etiologies and pathophysiological mechanisms. While the authors adopt a network approach, they provide insufficient justification for merging these cohorts, potentially oversimplifying divergent disease mechanisms. Separate analyses may offer a more nuanced understanding of tremor networks across these disorders.

Clarification of Clinical Relevance

The reported connectivity-outcome correlations are low to moderate in strength, raising concerns about their clinical significance. The authors should address whether these correlations meaningfully predict therapeutic outcomes and whether the modest effect sizes might limit the practical utility of their findings.

Lack of Significant Findings from Atrophy Maps

The study notes the absence of significant associations between tremor outcomes and atrophy maps. This raises questions about the role of structural degeneration in tremor pathophysiology and whether its exclusion would impact the multimodal network proposed. An explanation of these results, as well as their implications for the study's conclusions, is needed.

We thank the reviewer for this nice summary and hope we could address all points satisfactorily.

Reviewer #3

Remarks to the Author:

In their work, Goede et al. aim to explore whether a network of brain regions is associated with the multifaceted manifestations of tremor. By integrating results from lesion-based studies, functional imaging, and research involving patients with implanted DBS leads, the authors seek to create a unified representation of a tremor network. While the study addresses an important and underexplored area, several aspects warrant further discussion or clarification.

The reported findings of a network involving the primary motor area (M1), the supplementary motor area (SMA), and the cerebellum are unsurprising and largely reiterate what is already well documented in the existing literature. However, particularly interesting is the demonstrated opportunity to compare the two cohorts using functional lesion-based methods at different sites. At this point, the group might clarify why less than 10% of the variance (R^2 , assuming this is a correlation coefficient) is explained, despite a visually strong agreement. This aspect was rather unexpected.

Thank you for this important question. While the observed effect sizes may appear low, it is essential to emphasize that clinical outcomes are influenced by numerous factors beyond electrode location (e.g. patient age, comorbidities, baseline symptom severities, test-retest reliability in scores, imaging inaccuracy, etc.). Critically, however, electrode localization is one of the few factors that can be influenced, clinically, so it is of critical importance to medical care. As noted in our discussion, we accounted for heterogeneity by integrating multimodal data across DBS targets and performing separate analyses to clarify both shared and target-specific network components. To our knowledge, this represents the first effort to combine lesion mapping, atrophy patterns, EMG-fMRI, and DBS data across three different DBS targets, resulting in a robust and generalizable model.

To address the limitations of focusing solely on electrode location, we conducted additional analyses exploring factors such as patient-specific characteristics and baseline scores. These findings are presented in Fig. S1, providing further insights into the multifactorial determinants of DBS efficacy. Based on this analysis, we would expect to explain a maximum of ~10-30% of variance based on electrode localization alone, at least roughly matching our results.

We added a paragraph in the limitations section of our manuscript:

“While our linear regression model accounted for cohort differences and their interaction with connectivity, it is important to note that the model explained only a modest portion of the variance in clinical improvement (adjusted $R^2 = 0.13$ for the lesion derived map and $R^2 = 0.13$ for the EMG-fMRI derived map). This suggests that other unmeasured factors, such as individual patient characteristics, differences in disease pathology, or variability in clinical scales, may play a significant role. We approximate the impact of these factors according to literature reports in Fig. S1. Additionally, the variability in clinical scales (e.g., MDS-UPDRS-III for Parkinson’s disease vs. FTM-Tremor scale for essential tremor) may influence the sensitivity of our analysis and should be considered when interpreting these results. Critically, however, none of these variables may be influenced due to medical practice, with the sole exception of the electrode placement and stimulation settings, which renders the defined tremor response target (which is based on these two factors alone) the critical anchor point with an opportunity to potentially improve patient care ²⁵.” – discussion, p. 13

Fig. S1. Modeling considerations. The pie charts depict factors influencing patient outcomes following DBS, as determined through literature review. For Parkinson's disease (PD), more studies have investigated determinants of DBS outcomes compared to other conditions. Key considerations include the variance that DBS modeling can realistically explain. Clinical improvements depend on multiple factors beyond electrode placement and stimulation volumes, such as disease subtype, age, sex, levodopa response, disease duration, and comorbidities, which collectively account for approximately 50% of the variance in DBS response. Measurement noise, including inter- and intra-rater reliability of UPDRS-III scores, contributes another ~20%. Imaging resolution, electrode placement inaccuracies, and modeling limitations add ~10% variance in PD and ~30% in essential tremor (ET). Habituation effects are present particularly in VIM-DBS for ET. Further factors remain uncertain but may further influence outcomes. Despite these challenges, such models remain clinically valuable. Electrode placement and stimulation settings are modifiable, unlike immutable patient factors such as age or disease type. Consequently, identifying an optimal target that explains even ~10% of variance represents a meaningful advancement in the field¹⁻¹⁰.

*It is uncertain how much variance in explained clinical outcomes imaging reconstruction errors will have since no study deliberately tested for this. However, multiple reports speak of reconstruction errors of at least 1 mm displacement⁴⁻⁶. The only study that we are aware of which tested for artificially introduced spatial uncertainty imposed a jitter of N=258 electrode placements according to a 3D Gaussian distribution of 2mm full width half maximum. In this report, the standard error interval in outcome predictions for iteratively jittered group analyses ranged from ~60 % and ~70 % explained variance, i.e. a standard error of around 10%.

I believe the findings could be better contextualised if the clinical data were described more clearly. The tremor scores do not seem widely recognised.

We thank the reviewer for bringing this to our attention. For this purpose, we added a paragraph to the methods section to further describe the used scores, that represent widely used, and established scales:

"To measure tremor severity and its changes, we employed validated scales for ET and PD. A second ET cohort of patients that underwent VIM-DBS at Jacksonville, Florida¹² served as a validation cohort. For ET, we used the Fahn-Tolosa-Marin (FTM⁵⁴) Tremor Rating Scale, while for PD, we used the MDS-UPDRS Part III⁵⁵. Subscales assessed rest, postural, and action tremor on a 5-point scale (0-4) for each hand. In the Berlin, and Amsterdam PD cohorts, the original UPDRS-III version was used, providing a combined score for action and postural tremor. Also, the constancy of rest tremor was not consistently available across cohorts." – methods, p. 15

What was the rationale for not summing all tremor-related items of the UPDRS (MDS-UPDRS-III, Items 3.15-3.18) and just using action and rest tremor scores?

Thank you for your question and are sorry for the confusion. To clarify: All tremor-related items of the UPDRS have been used. We are sorry this was not clear and we now describe this more clearly in our manuscript.

However, to further explore this, we included a supplementary figure comparing the optimal response networks for rest tremor in Parkinson's disease (PD) patients with the combined action- and posture-tremor scores in both PD and essential tremor (ET) patients. However, our mapping approach could not reliably analyze rest tremor in ET patients due to its low prevalence in the cohort (mean rest tremor score < 0.5 points).

Fig. S7. Comparison of tremor subtypes. Correlation maps representing networks for optimal clinical response demonstrate similarities across DBS target regions, disease types, and tremor subtypes (rest, action, and posture tremor). The maps highlight overlapping areas of connectivity associated with effective tremor suppression, suggesting a shared therapeutic network underlying tremor relief irrespective of these variables.

This analysis was not initially performed due to the clinical rationale that, for example, tremor-dominant PD patients may be implanted in the VIM regardless of tremor type, as both rest and action tremors typically respond well to VIM-DBS with the same stimulation settings. Therefore, we did not assume that specific DBS settings would favor one tremor type over another, a perspective consistent with the clinical experience of our tremor expert coauthors.

We further added a paragraph to the methods section describing the used scores:

"To measure tremor severity and its changes, we employed validated scales for ET and PD. A second ET cohort of patients that underwent VIM-DBS at Jacksonville, Florida¹² served as a validation cohort. For ET, we used the Fahn-Tolosa-Marin (FTM⁵⁴) Tremor Rating Scale, while for PD, we used the MDS-UPDRS Part III⁵⁵. Subscales assessed rest, postural, and action tremor on a 5-point scale (0–4) for each hand. In the Berlin, and Amsterdam PD cohorts, the original UPDRS-III version was used, providing a combined score for action and postural tremor. Also, the constancy of rest tremor was not consistently available across cohorts." – methods, p. 15

According to what criteria were the leads implanted, and the targets chosen—particularly considering that the ventrolateral thalamus is highly heterogeneous and, in my experience, surgical approaches vary? Were there also cases where the region below the posterior ventrolateral thalamus was targeted in ET patients? The close anatomical relationship between the posterior subthalamic areas—possibly representing the dentatorubrothalamic tract (DRT)—and the STN in PD patients, as well as the potentially high energy levels possibly even in upper (dorsal) contacts, could lead to poor differentiation between the networks and blur the results by Goede et al.

Thank you for highlighting this important point. Blurring between targets (proximity of VIM, PSA and VIM) was of great concern to us, which is, for instance, why we developed a more highly resolved functional connectome deliberately for this study. We also intentionally selected a clinically representative target from a large DBS center with long-standing experience. However, we agree with the reviewer that these measures may still not completely rule out blurring of targets on a group level and we very much agree that surgical techniques differ across centers. Acknowledging the limitation of including data from only one center, we now included an additional ET cohort that underwent VIM-DBS at the Mayo Clinic Florida. Confirming the concern of the reviewer, in direct comparison, the Mayo Clinic cohort more deliberately targeted VIM (active contacts centered in the nucleus) while our original cohort from Berlin on average activated contacts centered around the PSA region. A lot has been written about differences between VIM and PSA, and some authors have strong opinions about superiority/inferiority of one of them. Our retrospective study is not powered or aiming to compare them, however, and personally, we err on the side of a shared network (in form of DRT) for both targets, as e.g. beautifully demonstrated in the meta-analysis by Nowacki et al. (2022 Ann. Neurol.) or also discussed in our recent review paper (Neudorfer et al. 2024 Neurotherapeutics).

Including the Mayo cohort still largely strengthened reproducibility of our findings since both cohorts, despite their surgical difference, independently point to the same network:

“A second ET cohort of patients that underwent VIM-DBS at Jacksonville, Florida ¹² served as a validation cohort.” – methods, p. 15

“To address the potential influence of varying surgical approaches, an optimal connectivity profile (*R-map*) was calculated using an additional ET cohort from Jacksonville, Florida. This cohort was compared to the ET cohort from Berlin to assess consistency. Importantly, the Florida cohort was not included in the primary analyses and served exclusively as an independent validation check.” – methods, p. 17

“Since surgical approaches for DBS in ET may vary between centers, the optimal connectivity profile from an additional cohort from the Mayo Clinic Florida, Jacksonville, Florida ¹⁶ was compared to the VIM-DBS cohort analyzed in this study from Berlin and demonstrated similar regions involved in the optimal connectivity profile (see Figs. S3 and S4).” – supplement, p. 12

Fig. S3. Correlation maps from different VIM-DBS cohorts demonstrate a similar optimal connectivity profile. **A:** Correlation map (R-map) from the VIM-DBS cohort in Berlin (left) compared to the R-map from the Florida cohort (right) shows a similar optimal connectivity profile, despite variations in surgical approaches. **B:** Active contacts (dots) from the left hemisphere illustrate these differences, viewed from the left side. The BigBrain Atlas is used as the anatomical backdrop ¹¹.

The use of thalamic nomenclature is another point of contention. The term "ventral intermediate nucleus (VIM)" has been questioned by some researchers, who consider it more of a transitional zone between afferent pallidal and cerebellar fibres rather than a distinct entity (for reference, see: <https://movementdisorders.onlinelibrary.wiley.com/doi/10.1002/mds.10136>). It is a bit surprising that the authors—some of whom have significantly contributed to prior work in this area—chose to adopt a less precise terminology I have never seen before, such as "ventrointermediate nucleus". Opting for an alternative nomenclature might provide a clearer representation of the anatomical structures under investigation.

Thank you for raising this important point. The senior author would like to take personal responsibility for the oversight and for using ventrointermediate nucleus – which has been adapted from Hassler's nomenclature (nucleus ventrointermedius). We have now changed the nomenclature to ventral intermediate nucleus – and kept some usage of the term given the wide clinical use of "VIM-DBS". As can be seen in our aforementioned recent review on the topic (Neudorfer et al., 2024 Neurotherapeutics), we very much care about the anatomical details in this region as the reviewer does. To address this critical aspect of precise anatomical positioning, we have added a discussion point about the imprecision of the general area and a supplementary figure that illustrates placements of the average active contacts within the anatomical vicinity of the thalamic nuclei for both cohorts.

"As a critical point for discussion, it should be emphasized that while we stick to the clinical convention of referring to VIM-DBS for the ET cohorts, the actual target region differs across centers and may not always

focus exclusively on the ventral intermediate nucleus, proper ³⁰. Indeed, many centers choose to predominantly activate contacts below the thalamus, in an anatomically complex region that hosts both grey and white matter structures, and which has sometimes been descriptively termed the posterior subthalamic area (PSA) ^{6,7}. This distinction can be seen across the two cohorts included here, where electrode placement in one centered around the VIM proper, while the second on the PSA (Figs. S3, S4). Moreover, even the concept of the VIM as a nucleus has been questioned by some researchers, who considered it more of a transitional zone between afferent pallidal and cerebellar fibers rather than a distinct entity ⁴⁴. Complicating matters further, the region covered by the nucleus has received different names by different anatomists and not all segregated it in the same fashion ⁶. Shedding light on this complex discussion goes beyond the scope of our article. However, analyzing both VIM-DBS cohorts included here separately showed highly similar optimal connectivity profiles, despite their differing average stimulation coordinates (Fig. S3, S4).” – discussion, p. 12f.

We further added two figures that will show the relationship of individual electrode placements i) to the VIM nucleus as defined by the DISTAL atlas (novel fig. S3) and ii) average contact centers and standard deviations to both Morel and DISTAL atlases (novel fig. S4).

Fig. S3. Correlation maps from different VIM-DBS cohorts demonstrate a similar optimal connectivity profile. **A:** Correlation map (R-map) from the VIM-DBS cohort in Berlin (left) compared to the R-map from the Florida cohort (right) shows a similar optimal connectivity profile, despite variations in surgical approaches. **B:** Active contacts (dots) from the left hemisphere illustrate these differences, viewed from the left side. The BigBrain Atlas is used as the anatomical backdrop ¹¹.

Fig. S4. Mean active contact standard deviation across VIM-DBS cohorts reveals distinct locations within the motor thalamus. The average active contacts (grey circle) and standard deviations (dashed circle line) are shown in light blue (Berlin) and light red (Florida). **A:** Outlines from DISTAL atlas ¹². *Abbreviations* (Hassler / Schaltenbrand & Wahren nomenclature): A.pr, Ncl. anterior principalis; Ce., Central nucleus; D.im.e, Dorsal intermediate nucleus, external part; D.im.i., Dorsal intermediate nucleus, internal part; D.sf, Dorsal superficial nucleus; La.M., Internal medullary lamina; M, Medial nucleus; V.c.i, Ventral caudal nucleus, internal part; V.im.i, Ventral intermediate nucleus, internal part; V.im.e, Ventral intermediate nucleus, external part; VL, Ventral lateral nucleus; V.o.a, Ventral oral anterior nucleus; V.o.p, Ventral oral posterior nucleus (VL.a, ventral lateral anterior nucleus); Z.im.e, Ncl. zentro-lateralis intermedius, external part; Z.im.i, Ncl. zentro-lateralis intermedius, internal part. The 7 Tesla MRI of the ex vivo human brain is used as the anatomical backdrop ¹³. **B:** Outlines from Morel atlas ¹⁴. *Abbreviations* (nomenclature by Hirai and Jones and adopted by Morel et al. ¹⁴): CeM, Central medial nucleus; CL, Central lateral nucleus; CM, Centromedian nucleus; LD, Lateral dorsal nucleus; MDmc, Mediodorsal nucleus, magnocellular part; MDpc, Mediodorsal nucleus, parvocellular part; Pf, Parafascicular nucleus; VLpd, Ventral lateral nucleus, posterior dorsal part; VLpv, Ventral lateral nucleus, posterior ventral part; VM, Ventral medial nucleus; VPM, Ventral posteromedial nucleus.

Convergent causal mapping of a tremor treatment network
(Goede et al.) – *Response to Referees*

Legend:

Reviewer Comment

Author Response

Changes in Manuscript

We would like to thank the reviewers for their thoughtful comments.

Reviewer #1

Remarks to the Author:

Appreciate the very thorough and comprehensive responses.

We are pleased that we were able to satisfactorily address all the points raised by the reviewer.

Reviewer #2

Remarks to the Author:

I have carefully reviewed the authors' responses and acknowledge that most concerns have been adequately addressed. However, one critical issue remains unresolved, which could challenge the premise that a common tremor network represents a superior therapeutic target compared to those based on pathophysiologic circuitry, such as basal ganglia targets for Parkinson's disease (PD) or rest tremor.

Specifically, an important question from the initial review remains unanswered: How does GPi targeting for PD tremor—despite being outside the cerebello-thalamo-cortical circuitry—yield better outcomes than VIM targeting (data presented in the manuscript), which is located within the hypothesized tremor network? Do these findings fully support the proposed conclusions?

Thank you for raising this critical question. We fully acknowledge the apparent paradox that GPi stimulation, despite being outside the cerebello-thalamo-cortical circuitry, may yield effective tremor outcomes. We must emphasize that our trial by no means should be used to compare effect sizes, however, since it involves retrospective data based on different surgeons / centers rather than a head-to-head prospective clinical trial. So, while we agree with the reviewer that the seeming paradox is important, we do not believe GPi leads to superior effects, which is also not the common understanding in the field.

We have now devoted an extensive discussion of this topic, outlining the key thoughts and hypothesis that may explain therapeutic effects of GPi-DBS in Parkinsonian tremor. We included a new analysis that showed that voxels in the posterior GPi (precisely matching the therapeutic target), are most strongly connected to the hypothesized tremor network and added additional references and a novel supplementary table as follows:

“S1: Therapeutic effects of GPi-DBS in Parkinsonian tremor

A key question of our results is how GPi-DBS (as used to treat PD tremor) yields effective outcomes despite being outside the cerebello-thalamo-cortical circuitry that we identify as the main surrogate of tremor effects based on multi-modal data in the present manuscript. The following thoughts may help conceptualize this seeming paradox:

1. GPi as a "switch" for tremor episodes

Recent models suggest that the GPi may act as a trigger for Parkinsonian tremor episodes, while the cerebello-thalamo-cortical circuit serves as a “dimmer” controlling tremor amplitude, as already discussed in the main text of this manuscript (see discussion and ²²). This aligns with previous studies showing that STN stimulation, which also influences GPi, can effectively reduce tremor. The ability of GPi-DBS to suppress tremor may thus stem from its role in modulating the entire tremor-generating network rather than acting at a single focal target. While an empirical analysis demonstrating this relationship is not possible using methods of the present manuscript, we aimed at estimating the influence of voxels within the pallidal region onto the tremor network by calculating each voxel's normative connectivity profile and spatially comparing its similarity to the tremor network. This led to peaks in voxels which had a connectivity profile most closely matching the profile of the convergent tremor network identified here. This analysis showed that the voxels with the strongest similarity precisely matched the posterolateral GPi region (matching the DBS target used in PD; figure S12).

Fig. S12. Spatial correlation between connectivity profiles of all pallidal voxels to the multimodally informed convergent tremor network. For each voxel in the subcortical field of view, functional connectivity was seeded and averaged across 1,000 rs-fMRI scans in the normative HCP connectome. These profiles were then compared to the convergent tremor network (fig. 5) using spatial correlations. High correlations represent voxels whose connectivity profile matches the spatial properties of the multimodal tremor network more precisely than others. Within the pallidum, this analysis peaked within its dorsolateral sensorimotor region, precisely matching the typical location of DBS electrodes implanted to treat PD symptoms (including tremor). While indirect, this analysis may support the notion that GPi may function as a polysynaptic ‘switch’ to modulate the tremor network as originally hypothesized in ²².

2. Network interactions between the basal ganglia and cerebello-thalamo-cortical circuit

Parkinsonian tremor differs fundamentally from other forms of tremor (e.g., essential tremor) in that it originates from pathological oscillations within the basal ganglia, which secondarily modulate the cerebello-thalamo-cortical circuit. This suggests that targeting GPi, an upstream nucleus within the basal ganglia-thalamo-cortical loop, can interrupt aberrant tremor activity before it propagates downstream. This is consistent with prior work showing that GPi acts as a key relay in the tremor circuitry, influencing both the basal ganglia and cerebello-thalamo-cortical pathways ³².

3. Tremor cells in the GPi and their role in parkinsonian tremor

Electrophysiological recordings have identified tremor cells within the lower portion of GPi, with activity that is highly correlated with tremor frequency in Parkinson’s disease ³². This suggests that GPi may play a direct role in tremor generation in PD, in contrast to essential tremor, which is primarily driven by cerebellothalamic dysfunction. Furthermore, cerebral activity related to Parkinson’s tremor has been shown

to first arise in the GPi and then propagate to the cerebello-thalamo-cortical circuit, suggesting that GPi stimulation might disrupt this pathological loop at its origin ²².

4. Influence of dopamine on GPi and tremor suppression

Dopamine is known to reduce tremor-related activity in both thalamus and GPi. In Parkinson's disease, dopaminergic depletion leads to pathological synchrony within the GPi, contributing to the emergence of resting tremor ²⁸. The fact that GPi-DBS improves tremor suggests that it may restore normal basal ganglia output and thereby indirectly modulate downstream (cerebellothalamic) tremor circuits.

5. The role of diaschisis and lesion network mapping (LNM) in understanding GPi effects

An additional parallel to understanding remote network effects of specific stimulation sites may be derived from results in the lesion network mapping (LNM) literature. Traditional lesion studies have demonstrated that symptoms such as tremor ³³, hemichorea ³⁴, or parkinsonism ³⁵ do not always localize to a single anatomical region but rather to a distributed functional network. Even when lesion locations vary, they often map onto the same functionally connected network via the phenomenon of diaschisis, the remote impact of a lesion on connected brain regions ³⁶.

In our case, GPi-DBS may leverage similar mechanisms, exerting network-level effects despite being anatomically distinct from the cerebello-thalamo-cortical tremor circuit. This aligns with prior LNM studies showing that lesions in different brain locations can converge onto a shared symptom-specific network (for examples, see table S2). If a lesion anywhere within a tremor network may cause tremor, then neuromodulation of a functionally connected region, such as the GPi, may similarly alleviate tremor, even if it is not the direct site of pathophysiology." – supplementary material

The following paragraph was added to our discussion:

"Our results further indicate that targets outside the classic cerebello-thalamo-cortical circuitry, such as the GPi, can achieve tremor suppression by modulating network-level dynamics. This reinforces the concept that effective neuromodulation depends on the integration of multiple neural circuits rather than a single anatomically defined region. The principle of lesion network mapping provides a useful analogy, as it demonstrates that functionally connected regions, rather than isolated anatomical sites, drive symptom expression. In the case of Parkinsonian tremor, GPi and the basal ganglia circuitry is already seen as a 'switch' that may influence the cerebello-thalamo-cortical circuit indirectly through polysynaptic pathways. We have devoted an extensive supplementary section including additional analyses and literature reviews on this important topic in section S1." – discussion, p. 12

The table S2 was added to the supplementary material, p. 16:

Supplementary table 2 (table S2): Lesion network mapping (LNM) studies in motor symptoms: Summary of studies demonstrating how lesions in different brain regions can map onto common brain circuits.

Motor symptom	N	Description
Cervical dystonia ³²	25	Lesions inducing cervical dystonia showed positive connectivity to the cerebellum and negative connectivity to the somatosensory cortex.
Freezing of gait ³³	14	Anatomically heterogenic lesions showed in 13/14 cases network overlap in the dorsal medial cerebellum.
Hemichorea ²⁹	29	Anatomically heterogenic lesions showed in 90% of the cases network overlap in the posterolateral putamen.
Holmes' tremor ³⁴	26	Cases were connected to a common brain circuit with nodes in the red nucleus, thalamus, globus pallidus, and cerebellum.
Parkinsonism ³⁰	29	Most sensitive and specific connectivity was to the claustrum. 31 % of lesions hit the substantia nigra.
Tics ³⁵	22	Lesions mapped to a common brain circuit involving insular cortices, cingulate gyrus, striatum, globus pallidus internus, thalami and cerebellum. Connectivity to the putamen was specific to tic-inducing lesions.

We hope this response clarifies the network-based rationale for the efficacy of GPi-DBS and why these findings do not contradict but rather reinforce our proposed conclusions.

Reviewer #3

Remarks to the Author:

Goede and his co-authors put an impressive amount of work into their revision of the manuscript now titled 'Convergent Mapping of a Tremor Treatment Network' and attempted to address the numerous questions raised by the three reviewers. Of particular note is the additional cohort treated at the Mayo Clinic Florida. One inevitably wonders why this cohort was used 'only' for validation and why the primary analyses were not supplemented. Looking at the active contacts (Figures S3/S4), they are very different. This raises the question of whether this difference affects the credibility of the results, as it seems almost irrelevant at what point the THS impulses arrive to activate the networks. Given the small effects observed, this is, of course, an important limitation to the overall results.

Thank you for raising these important points. During the revision process, we were kindly allowed to include additional data from the Mayo Clinic Florida cohort, which enabled us to address key concerns raised in the initial round of review. Given the temporal order, we chose to use the Florida cohort to validate our results, which we deem a much stronger concept than just merging cohorts together. Namely, this shows that the map we first created (and which reviewers have already seen) would generalize to an unseen hold-out cohort.

This being said, we have now gladly included a reanalysis of the entire cohort together. When repeating all analyses from the main manuscript using both cohorts combined (instead of Berlin alone) all results remained significant and highly similar (as summarized by novel figure S6):

Fig. S6. Replication of the main analysis including the Florida VIM-DBS cohort. This analysis included N = 36 VIM-DBS patients from Berlin, Germany (bilateral DBS), N = 19 VIM-DBS patients from Florida, USA (unilateral DBS), and all STN-DBS patients from the main analysis. VIM patients are represented in blue, STN patients in orange. As a limitation, in the Florida cohort, only FTM sum scores were available, whereas the main analysis used the sum of specific FTM tremor score items to best match MDS-UPDRS-III tremor scores. **A:** Positive correlation between relative clinical improvement following DBS and each electrode's connectivity to the lesion network map and the fMRI-derived map (original analyses in figures 1 and 2) **(B).** **C:** Connectivity correlation map (R-map) derived from the full VIM-DBS cohort, including patients from both Berlin and Florida, as used in the replication analysis (original figure 3). This map represents the optimal connectivity profile associated with clinical improvement across the expanded VIM-DBS dataset. **D:** VIM-DBS electrodes' VTA connectivity to the STN cohort's optimal connectivity profile (R-map) correlated with clinical improvement (original figure 3). **E:** Agreement map (left) combining STN and VIM cohorts when including both Berlin and Florida cohorts. Right panel: positive correlation between DBS site

connectivity to the agreement map and clinical improvement (original figure 4). **F:** Multimodally-informed convergent tremor map, created by overlaying the lesion-derived map, EMG-fMRI-derived map, VIM-STN agreement map when including both Berlin and Florida cohorts, and an ET-specific atrophy map, with z-scores visualized. In an out-of-sample cohort of 31 PD hemispheres with GPI-DBS, connectivity to this map correlated with MDS-UPDRS-III tremor improvement (original figure 5). A cohort regressor was applied in all analyses to account for potential confounding effects related to differences between patient groups.

The reviewer correctly notes that electrode placements across the two cohorts differ. In our view, this is a key strength of our results (especially when using the first cohort to build the network and then being able to explain variance in the unseen cohort that was targeted differently). The two cohorts are mainly different in the z-axis, conforming to the course of the cerebellothalamic pathway flowing into the motor thalamus. Indeed, this exact relationship has been shown multiple times in the recent past, with published optimal DBS sweetspots following the same exact trajectory. We summarize this relationship in novel figure S3:

Fig. S3. Optimal VIM-DBS contacts align along the cerebellothalamic pathway rather than to an anatomical sweet spot. Accumulating evidence from the present study and recent publications. A: Active contacts of all VIM-DBS electrodes from the present study, including those from the Berlin cohort (light blue) and the validation Florida cohort (dark blue), aligned with the trajectory of the cerebellothalamic pathway / dentatorubrothalamic tract (DRT, purple and green). DRT fibers are shown as defined by the Basal Ganglia Pathway Atlas ¹¹. **B:** Medial view of the thalamus and posterior subthalamic area (PSA), showing previously reported target coordinates associated with symptom improvement. Sites linked to better clinical outcomes are closely related to the cerebellothalamic outflow tract (dark red) and the zona incerta (white mesh) at the level of the PSA. Panel adapted, with permission, from ¹². **C:** Medial view showing an excellent responder cluster from the largest meta-analysis to date in VIM-DBS for tremor, as recently identified. Panel adapted, with permission, from ¹³. **D:** The white sphere represents the classic VIM target at the ventral border of the VIM nucleus (left side of the brain in posterior view), while recently recommended targets from the literature are shown on the right side of the brain in posterior view. The VIM nucleus is shown in yellow, and the DRT is displayed in blue. Panel adapted, with permission, from ¹⁴. **E:** The dentatothalamic tract is shown in blue, alongside recently published sweet spots for optimal tremor control. Panel adapted, with permission, from ¹⁵.

We appreciate the opportunity to clarify these points and believe that the inclusion of the Florida cohort not only adds external validation but also reinforces the robustness of our network-based approach.

Above and beyond the two new figures, the following paragraphs were added:

“Consistently, we found that the active contacts of all VIM-DBS electrodes, including those from both the Berlin cohort and the Florida cohort, aligned with the trajectory of the DRT, reinforcing its relevance as a key anatomical substrate for effective tremor control (see Fig. S3).” – discussion, p. 13

“Importantly, results remained overall consistent when including both cohorts in a supplementary replication of the main analysis (Fig. S6), reinforcing the robustness of the identified tremor network.” – discussion, p. 13

Convergent mapping of a tremor treatment network
(Goede et al.) – Response to Referees

Legend:

Reviewer Comment

Author Response

Changes in Manuscript

Reviewer #2

Remarks to the Author:

I would like to once again commend you on this outstanding work—one that very few research groups could currently produce with such depth and rigor. The organization, data collection from multiple centers, integration of lesion studies, multimodal data processing, and comprehensive discussion make this study an exceptional contribution to the field. The integration of diverse data sources (MRI, connectomics, lesion studies) not only strengthens hypotheses related to DBS targeting but also offers crucial insights into circuit-level mechanisms underlying neurological symptoms.

Your work convincingly demonstrates, through robust data, that tremor across different etiologies is fundamentally linked to abnormal oscillatory activity within the cerebello-thalamic circuit, which appears essential for its manifestation. The manuscript effectively illustrates how different tremor types converge on a shared neural network, regardless of whether their initial pathological origin lies within adjacent circuits such as the basal ganglia (e.g., Parkinsonian tremor).

However, while the study establishes a strong case for a convergent tremor network, it is less conclusive in defining it as a universal therapeutic target. The success of GPi DBS—despite targeting a structure outside the cerebello-thalamic circuit—suggests that tremor treatment does not necessarily require direct modulation of this network. Instead, pathological activity may originate externally but propagate through the cerebello-thalamic loop, where it manifests as tremor. Similarly, this may be the case for STN DBS, another well-established intervention for Parkinsonian tremor. Given that both GPi and STN play key roles in tremor generation, their omission raises questions about whether the proposed "Convergent Mapping of a Tremor Treatment Network" fully accounts for treatment effectiveness. Additionally, GPi and STN DBS provide broader benefits beyond tremor control, improving other Parkinsonian symptoms—an advantage VIM-DBS does not achieve to the same extent.

In summary, while your study convincingly supports a common tremor network, it does not fully establish it as a universal therapeutic target, as effective interventions may act on external circuits rather than the tremor network itself. This principle may extend to other tremor types, such as dystonic and Holmes tremor, further emphasizing the need for tailored therapeutic approaches.

Nevertheless, this manuscript undoubtedly deserves acceptance for publication. Future research should continue exploring novel and more effective DBS targets to enhance tremor control, minimize adverse effects, and improve overall patient outcomes. Clarifying these distinctions would further strengthen the manuscript's impact and its contribution to the field.

We are pleased to have satisfactorily addressed all points raised by the reviewer.

Reviewer #3

Remarks to the Author:

NO further comments.

We thank the reviewer for the constructive feedback during the revision process and are pleased that all comments and requests have been satisfactorily addressed.